# Training confounder-free deep learning models for medical applications

Qingyu Zhao 1,4, Ehsan Adeli 1,2,4 & Kilian M. Pohl 1,3✉

The presence of confounding effects (or biases) is one of the most critical challenges in using deep learning to advance discovery in medical imaging studies. Confounders affect the relationship between input data (e.g., brain MRIs) and output variables (e.g., diagnosis). Improper modeling of those relationships often results in spurious and biased associations. Traditional machine learning and statistical models minimize the impact of confounders by, for example, matching data sets, stratifying data, or residualizing imaging measurements. Alternative strategies are needed for state-of-the-art deep learning models that use end-to-end training to automatically extract informative features from large set of images. In this article, we introduce an end-to-end approach for deriving features invariant to confounding factors while accounting for intrinsic correlations between the confounder(s) and prediction outcome. The method does so by exploiting concepts from traditional statistical methods and recent fair machine learning schemes. We evaluate the method on predicting the diagnosis of HIV solely from Magnetic Resonance Images (MRIs), identifying morphological sex differences in adolescence from those of the National Consortium on Alcohol and Neurodevelopment in Adolescence (NCANDA), and determining the bone age from X-ray images of children. The results show that our method can accurately predict while reducing biases associated with confounders. The code is available at https://github.com/qingyuzhao/br-net.

---

[1] Department of Psychiatry & Behavioral Sciences, Stanford University, Stanford, CA 94305, USA. [2] Department of Computer Science, Stanford University, Stanford, CA 94305, USA. [3] Center for Biomedical Sciences, SRI International, Menlo Park, CA 94205, USA. [4] These authors contributed equally: Qingyu Zhao, Ehsan Adeli. ✉email: kilian.pohl@stanford.edu

A fundamental challenge in medical studies is to accurately model confounding variables[1–3]. Confounders are extraneous variables that distort the apparent relationship between input (independent) and output (dependent) variables and hence lead to erroneous conclusions[4–6] (see Fig. 1). For instance, when neuroimaging studies aim to distinguish healthy individuals (a.k.a controls) from subjects impacted by a neurological disease, the input variables are images or image-derived features, and the output variables are the class labels (i.e., diagnosis). If the average age of the diseased cohort is significantly older than the healthy controls, the age of individuals potentially confounds the study[7–9]. When not properly modeled, a predictor may learn the spurious associations and influences created by the confounder (age, in this case) instead of the actual biomarkers of the disease[10].

Traditionally, studies control for the impact of confounding variables by eliminating their influences on either the output or the input variables. With respect to the output variables, one can reduce the dependency to confounders by matching confounding variables across cohorts (during data collection)[7] or through analytical approaches, such as standardization and stratification[11,6]. Associations between confounders and input variables are frequently removed by regression analysis[12,6], which produces residualized variables that are regarded as the confounder-free input to the prediction models.

The most advanced image-based prediction models are based on convolutional neural networks (ConvNets)[13,1,14–16,3]. A standard ConvNet contains a feature extractor ($\mathbb{FE}$) followed by a classifier/predictor ($\mathbb{P}$). $\mathbb{FE}$ reduces each medical image to a vector of feature $\mathbf{F}$, based on which the fully connected layers of $\mathbb{P}$ predict a binary or continuous outcome $\mathbf{y}$ (Fig. 2a). Unlike traditional machine-learning models, ConvNets require large training data sets and adopt end-to-end learning strategy to extract feature $\mathbf{F}$ on-the-fly from the raw image $\mathbf{X}$. This renders the above methods to account for confounders unsuitable as they either result in reduced number of training samples (e.g., matching or stratification) or require deterministic features that are computed beforehand (e.g., standardization or regression). Possible alternatives could be unbiased[17–21] and invariant feature-learning approaches[22–25] relying on end-to-end training to study the invariance (independence) between the learned feature $\mathbf{F}$ and a bias factor (① in Fig. 2b). Despite the similarity in the problem setup, ignored by these methods yet of great importance to medical imaging studies is selecting feature $\mathbf{F}$ predictive of the outcome $\mathbf{y}$ (i.e., ③ in Fig. 2b), while accounting for the intrinsic relationship between $\mathbf{y}$ and the confounder $\mathbf{c}$ (i.e., ② in Fig. 2b). An example of such an intrinsic relationship with respect to the age-confounded magnetic resonance imaging (MRI) dataset is to distinguish the healthy aging of the brain in controls from aging accelerated by a disease, such as HIV infection[26–28]. This paper proposes to account for this relationship by introducing the

learning scheme confounder-free neural network (CF-Net, Fig. 2a).

CF-Net exploits concepts from traditional statistical modeling within an invariant feature-learning scheme. Inspired by our technical report[22], we attach a lightweight component $\mathbb{CP}$ to $\mathbf{F}$, which quantifies the statistical dependency between $\mathbf{F}$ and $\mathbf{c}$ in order to guide $\mathbb{FE}$ in removing confounding effects in the feature-extraction process (Fig. 2a). The guidance is based on training the CF-Net via the min–max game as done by generative adversarial networks (GANs)[29]. In this iterative training process, $\mathbb{CP}$ aims to predict the value $\mathbf{c}$ from $\mathbf{F}$, $\mathbb{FE}$ aims to adversarially increase its prediction loss, and $\mathbb{P}$ aims to predict $\mathbf{y}$ based on the confounder-free features. Instead of enforcing marginal independence between $\mathbf{F}$ and $\mathbf{c}$ as we propose in ref. [22], a more principled way of correcting confounding effects is to only remove the direct association between $\mathbf{F}$ and $\mathbf{c}$ (① in Fig. 2b) while preserving their indirect association with respect to $\mathbf{y}$ (② and ③ in Fig. 2b). We therefore specifically train $\mathbb{CP}$ on a $\mathbf{y}$-conditioned cohort, i.e., samples of the training data whose $\mathbf{y}$ values are confined to a specific range (referenced as $\rho$ in Fig. 2a). In doing so, the features learned by CF-Net are predictive of $\mathbf{y}$ while being conditionally independent of $\mathbf{c}$ ($\mathbf{F} \perp\!\!\!\perp \mathbf{c} | \mathbf{y}$). We refer to this condition as confounder-free training. In the HIV example, CF-Net would learn to separate healthy controls ($\mathbf{y} = 0$) from HIV-positive patients ($\mathbf{y} = 1$) by training $\mathbb{CP}$ only on the control group to correctly model the normal aging effects of the brain captured by MRI. This is one of the first attempts to design an end-to-end, confounder-free prediction model for medical images, in which the goal is not only to learn features invariant to a bias variable, but also to properly model interactions among all three variables in a confounded situation.

We underline the utility of the proposed CF-Net by deploying it to predict HIV diagnosis from brain MRIs of adults ($N = 345$) that are confounded by age, identify sex differences in brain MRIs of adolescents of NCANDA ($N = 674$, age 12–21 years) with pubertal development as the confounder, and determine the bone age of children based on X-ray images of their hands ($N = 12{,}611$), where the cohort was confounded by gender. Through these experiments, we show the impact of $\mathbb{CP}$ on reducing the risk of deriving features and predictions affected by confounders. Beyond that, the supplement summarizes additional experiments on the three data sets and on a synthetic dataset. These results converge with the theoretical advantages of our adversarial loss function (over state-of-the-art-invariant feature-learning schemes). As we systematically studied in the technical report[22], these advantages include the ability to handle continuous confounding variables and guaranteeing mean independence between $\mathbf{F}$ and $\mathbf{c}$.

## Results

**HIV diagnosis from MRIs.** We applied CF-Net and a standard ConvNet (without $\mathbb{CP}$) to distinguish the T1-weighted brain MRIs of healthy controls ($N = 223$, age $45 \pm 17$ years) from patients ($N = 122$, age $51 \pm 8.3$ years) diagnosed with HIV (CD4 count >100 $\frac{\text{cells}}{\mu\text{L}}$). As HIV subjects were generally older, age was the confounder of this study. The prediction accuracy of the models was determined via fivefold cross-validation. On the four folds used for training, two cohorts of equal size were generated by data augmentation (see "Methods" section) to ensure that the model would not bias predictions toward the larger cohort, i.e., the control cohort. We then confined the predictions of age by $\mathbb{CP}$ to the controls (i.e., the $\mathbf{y}$-conditioned cohort was defined by $\mathbf{y} = 0$). The prediction accuracy on the testing folds was measured by balanced accuracy (BAcc)[10] (to account for different numbers of testing subjects in each cohort), and precision and recall rates according to the uninformative operating

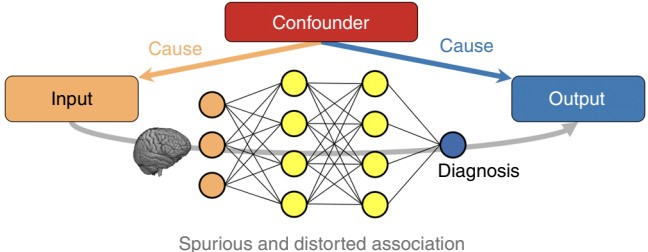

**Fig. 1 Confounding effects in deep-learning models.** A confounder is a variable that influences both the input and the output of a study causing spurious association, if not properly controlled for.

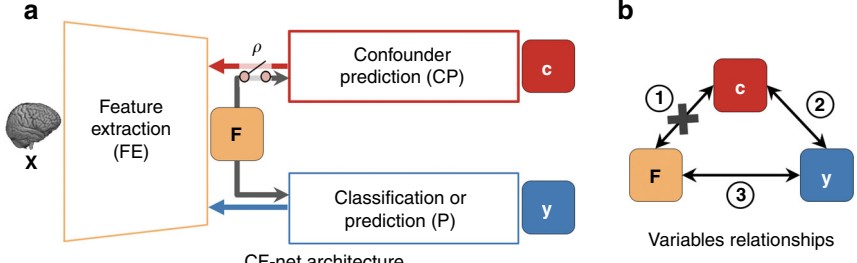

**Fig. 2 The proposed confounder-free neural network (CF-Net). a** Model architecture for confounder-free feature learning, composed of three subnetworks. $\mathbb{FE}$ learns features that successfully predict ($\mathbb{P}$) the outcome **y** while being invariant (conditional independence defined by $\rho$) to the confounding variables with the help of the adversarial component $\mathbb{CP}$. **b** The confounder **c** influences both the output **y** (i.e., ②) and the input **X**, from which feature **F** is extracted (i.e., ①). The classifier deems to find the relation ③ to enable prediction of the output labels. Our adversarial component aims to remove the direct dependency between **F** and **c** ①.

**Table 1 Balanced accuracy (%), precision (%), recall (%), and $F_1$ score of HIV-diagnosis prediction.**

| Method | Whole cohort | | | c-Independent subset | | c-Independent young | | c-Independent old | |
|---|---|---|---|---|---|---|---|---|---|
| | BAcc | Pre, Rec | $F_1$ score | BAcc | Pre, Rec | BAcc | Pre, Rec | BAcc | Pre, Rec |
| ConvNet | 71.6 | **78.2**, 59.8 | 0.68 | 68.4 | **84.4**, 52.5 | 59.7 | **85.0**, 36.3 | 75.3 | 85.0, 65.7 |
| CF-Net | **74.1** | 73.4, **75.4** | **0.74** | 74.2* | 73.0, **75.4** | 69.0* | 76.7, **62.7** | 82.4 | **88.1**, **76.4** |

*Denotes significant higher balanced accuracy than ConvNet by DeLong's test ($p < 0.05$).
Best results in each column are typeset in bold.

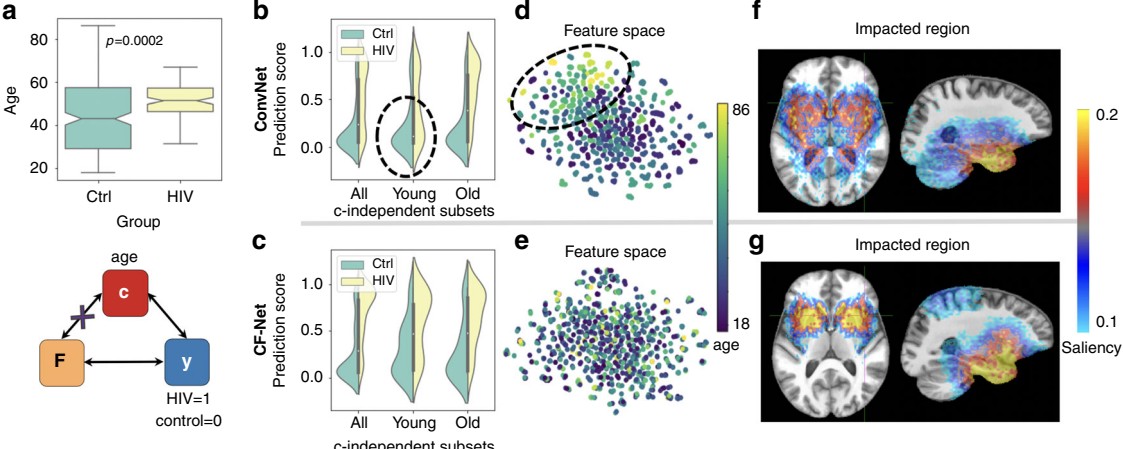

**Fig. 3 HIV diagnosis from MRIs. a** Age discrepancy ($p = 0.0002$, two-tailed two-sample $t$-test) between $n = 223$ control (Ctrl) subjects and $n = 122$ HIV patients resulted in the baseline ConvNet learning the confounding effects (**b, d, f**), which were alleviated by the proposed CF-Net (**c, e, g**). Boxplots are characterized by minimum, first quartile, median, third quartile, and maximum. **b, c** HIV-prediction scores measured on a subset of $n = 122$ control and $n = 122$ HIV subjects with the same age distribution (**c**-independent). **d, e** t-SNE visualization of the feature space learned by the deep-learning models. **f, g** Saliency maps[33] corresponding to the voxel-level attention (larger attention means more discriminative voxels) by the models.

point of 0.5. To investigate if the prediction of the models was confounded by age, we also recorded the three accuracy scores of the approaches (without retraining) on a confounder-independent subset (**c**-independent). The **c**-independent subset in this experiment was a subset of HIVs and controls with the same distribution of age (122 controls: $50.1 \pm 11.5$ years, 122 HIVs: $50.6 \pm 8.4$ years, $p = 0.9$ $t$-test).

CF-Net achieved a BAcc of 74.1% on the whole cohort, which was higher than the BAcc of ConvNet (71.6%) (Table 1). Although this improvement was only on a trend level according to DeLong's test (two-tailed $p = 0.068$), CF-Net recorded a more balanced precision (73.4%) and recall scores (75.4%) than ConvNet, which had a tendency to label subjects as controls

(low recall, Fig. 3b). Compared to the whole cohort, CF-Net recorded a similar BAcc of 74.2% on the **c**-independent subset, which was significantly higher (two-tailed $p = 0.035$, DeLong's test) than the BAcc of ConvNet (BAcc: 68.4%). Further, CF-Net recorded similar accuracy, precision, and recall on the **c**-independent subset, whereas the discrepancy between precision and recall for ConvNet further increased. Confining the computation of the accuracy score to the older and younger subjects (subcohorts divided by the mean age of 50.4 years) revealed that the predictions by ConvNet were largely biased by age. While both models recorded similar BAcc on the older cohort (two-tailed $p = 0.36$, DeLong's test), CF-Net was significantly more accurate than ConvNet on the younger cohort

(two-tailed $p = 0.045$, DeLong's test). As indicated by the black circles in Fig. 3b, most of the young HIV subjects were falsely labeled as controls by ConvNet (only 36.3% recall rate according to Table 1) as the control cohort was significantly younger than the HIV-positive subjects. On the other hand, CF-Net reduced the gap in prediction accuracy between the age groups (Fig. 3c).

To assess that the unbiased prediction of CF-Net was the result of extracting features impartial to normal aging, we performed a post hoc analysis, in which we trained $\mathbb{CP}$ to predict age from the learned features. Upon convergence of the training loss in each run of the fivefold cross-validation, the post hoc analysis retrained $\mathbb{CP}$ from scratch on the features extracted from the controls in the training folds and recorded the predicted age of the controls in the testing fold. According to Supplementary Fig. 2, the features learned by CF-Net no longer contained aging information as the prediction of age was nearly random (Pearson's $r = 0.12$, two-tailed $p = 0.17$). However, training $\mathbb{CP}$ on the features learned by 3D ConvNet resulted in age prediction of significant accuracy (Pearson's $r = 0.95$, two-tailed $p < 0.0001$). These results were also supported by measuring the statistical dependence between the features and age via distance correlation ($dcor$)[30] and mutual information (MI)[31]. Based on a bootstrapping analysis, CF-Net achieved an average of $dcor = 0.07$ and $MI = 0.02$, which were significantly lower (two-tailed $p < 0.001$, two-sample $t_{38} > 14.2$) than $dcor = 0.21$ and $MI = 0.13$ reported for ConvNet (Supplementary Fig. 3). We visually confirmed this finding by projecting the high-dimensional $\mathbf{F}$ of each control subject into 2D via t-SNE[32]. Figure 3d shows each subject as a point, whose color was defined by their age. While older subjects are concentrated on the upper-left region in the feature space associated with ConvNet, a clear pattern with respect to age was not visible for the projections associated with CF-Net (Fig. 3e).

To gain more insight into which anatomical regions drove the predictions, Fig. 3f, g visualizes the saliency maps[33] of ConvNet and CF-Net with yellow, highlighting areas that the predictions heavily relied upon. Figure 3f reveals that the ConvNet-extracted features close to the ventricles and cerebellum, which were crucial markers for brain aging[34] omitted by CF-Net. On the other hand, CF-Net produced higher saliency in the precentral and postcentral gyri, which are frequently linked to alternations in cortical structure and function in HIV-infected patients[35,36]. Other regions with high average saliency according to CF-Net are located in the temporal lobe, inferior frontal gyrus, and subcortical regions, including the amygdala and hippocampus. These regions (except for the amygdala) also exhibited significant white-matter tissue loss due to HIV according to a traditional voxel-based morphometry analysis[37] (Supplementary Fig. 4).

**Brain morphological sex differences in adolescent brains of the NCANDA study**. The public dataset (Release: NCANDA_PU-BLIC_BASE_STRUCTURAL_IMAGE_V01[38]) consisted of the baseline T1-weighted MRI of 334 boys and 340 age-matched girls (age 12–21 years, $p > 0.5$, two-sample $t$-test) from the National Consortium on Alcohol and NeuroDevelopment in Adolescence (NCANDA)[39] that met the no-to-low alcohol drinking criteria of the study. The confounder of the study was the pubertal development score (PDS, Fig. 4a)[39], which was significantly higher ($p < 0.001$, two-sample $t$-test) in girls ($3.41 \pm 0.6$) than boys ($2.86 \pm 0.7$).

With respect to the ConvNet baseline, the results from the previous experiment were largely replicated. Based on 5-fold cross-validation, the accuracy in predicting sex dropped from 90.3% across all samples to 87.3% (Table 2) on a $\mathbf{c}$-independent subset, which consisted of 200 boys and 200 girls with the same PDS distribution ($3.14 \pm 0.65$). Being significantly confounded by

PDS, the ConvNet produced a lower balanced accuracy (BAcc: 79.5%) for subjects in the early pubertal stage compared with an accuracy score of 90.6% for subjects in later stages (subcohorts divided by the mean PDS of 3.2). As boys had significantly lower PDS, the ConvNet tended to label girls with small PDS as boys (recall: 68.1%, Fig. 4b). Although the t-SNE projection of the ConvNet features showed less pronounced correlation with PDS compared with the HIV experiment (Fig. 4d), the confounding effect of PDS still significantly impacted the derived features as revealed by the post hoc training of $\mathbb{CP}$ (Pearson's $r = 0.84$, $p < 0.001$, Supplementary Fig. 7). Last, sex prediction of ConvNet was mostly based on the parietal inferior lobe, supramarginal region, cerebellum, and subcortical regions according to the saliency map of Fig. 4f (Supplementary Fig. 9).

For CF-Net, the accuracy depended on the set of subjects used for training the component $\mathbb{CP}$, which, unlike in the HIV experiment, was not uniquely defined as the modeling of the PDS effect that could be conditioned on $\mathbf{y} = 0$ (boys) or $\mathbf{y} = 1$ (girls). According to Table 2, conditioning the training of $\mathbb{CP}$ on boys resulted in more accurate predictions in the $\mathbf{c}$-independent subset and recorded a smaller gap in accuracy across subjects at different pubertal stages, while conditioning on girls not only reduced the BAcc, but also enlarged the discrepancy in precision and recall rates. As expected, similar degraded performance was also observed when training $\mathbb{CP}$ on subjects of both sexes without conditioning (CF-Net (All) in Table 2). Among the three implementations of CF-Net, only the CF-Net conditioned on boys was significantly more accurate in prediction at the early pubertal stage (two-tailed $p = 0.039$, DeLong's test) and produced features significantly less predictive of PDS ($p < 0.001$, one-sample $t_{333} = 12.2$, Supplementary Figs. 7 and 8) compared to ConvNet (Fig. 4b–e). Interestingly, the saliency map associated with this CF-Net implementation (Fig. 4f, g, Supplementary Fig. 9) focused only on subcortical regions.

**Bone-age prediction from hand X-ray images**. The dataset consisted of hand X-ray images of 12,611 children (6833 boys and 5778 girls) that were released by the Radiological Society of North America (RSNA) Radiology Informatics Committee (RIC) as a machine-learning challenge for predicting pediatric bone age[40]. The confounder in this study was sex as boys were significantly older than girls (boys: $134.8 \pm 42.2$ months, girls: $118.7 \pm 38.2$ months). We randomly chose 75% of the images ($N = 9458$) as training data and the remaining as validation data ($N = 3153$). The ConvNet was based on the publicly released implementation by the Kaggle challenge[41]. The feature extractor consisted of a pretrained VGG-16 backbone followed by an attention module[41]. This ConvNet achieved a mean absolute error of 13.8 months in predicting age from the X-rays of the validation set. The model tended to overestimate the age of girls compared to boys (Fig. 5b), and this discrepancy was more pronounced in the age range of 110–200 months (Fig. 5c).

Next, we aimed to remove sex-related confounding effects in the attention module by CF-Net. Since the ConvNet was based on a VGG-16 feature extractor pretrained on the large number of natural images provided by ImageNet, it was unlikely to contain confounding information for X-ray image. Hence, we only applied the $\mathbb{CP}$ component to adjust parameters of the attention module, but kept the VGG-16 feature extractor fixed. However, $\mathbf{y}$ was now a *continuous* variable as opposed to a binary one used in the previous experiments, so the $\mathbf{y}$-conditioned cohort could not be defined with respect to a fixed prediction outcome. Instead, we applied the $\mathbb{CP}$ component to a bootstrapped-training set of 10,000 boys and 10,000 girls whose age was confined to the interval from 75 to 175 months and had strictly matched

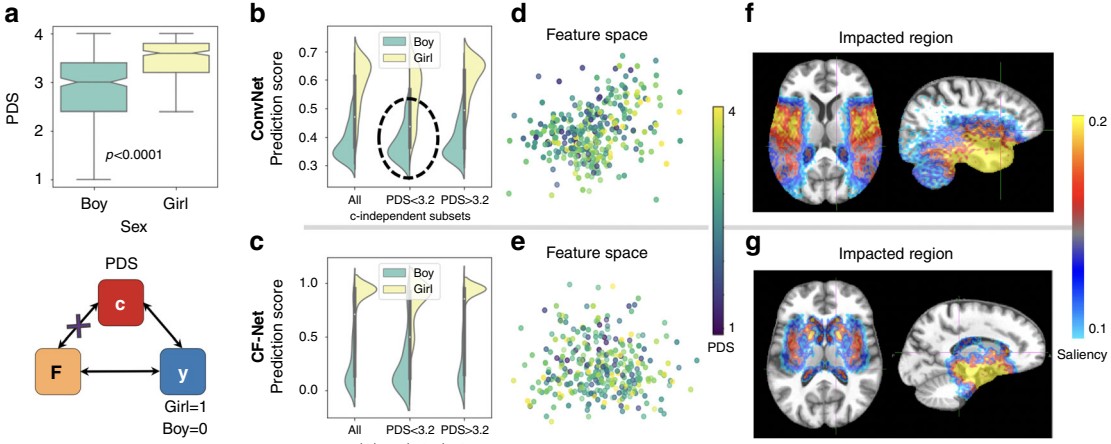

**Fig. 4 Sex prediction from adolescent brain MRIs. a** Significantly different pubertal development scores (PDS) between $n = 334$ boys and $n = 340$ girls ($p < 0.0001$ two-tailed two-sample $t$-test). Boxplots are characterized by minimum, first quartile, median, third quartile, and maximum. **b, c** Sex-prediction scores measured on all subjects and the **c**-independent subset containing $n = 200$ boys and $n = 200$ girls. **d, e** t-SNE visualization of the feature space learned by the deep-learning models. **f, g** Saliency maps of sex differences.

**Table 2 BAcc (precision and recall) on predicting sex from MRIs of NCANDA matched with respect to PDS. Optimal results were achieved when conditioning CF-Net on boys.**

| Method | Whole cohort | | | c-Independent | | PDS < 3.2 | | PDS > 3.2 | |
|---|---|---|---|---|---|---|---|---|---|
| | BAcc | Pre, Rec | $F_1$ score | BAcc | Pre, Rec | BAcc | Pre, Rec | BAcc | Pre, Rec |
| ConvNet | 90.3 | 95.5, 85.2 | 90.5 | 87.3 | 92.5, 82.5 | 79.5 | 92.8, 68.1 | 90.6 | 91.0, 90.0 |
| CF-Net (All) | 83.0 | 73.8, 92.2 | 82.0 | 83.3 | 93.0, 73.5 | 74.1 | 92.7, 56.5 | 87.3 | 93.1, 82.4 |
| CF-Net ($\mathbf{y} = 1$) | 85.2 | 72.1, 98.5 | 83.3 | 84.3 | 96.5, 72.0 | 78.2 | 98.6, 58.0 | 89.0 | 98.6, 79.4 |
| CF-Net ($\mathbf{y} = 0$) | 88.8 | 93.6, 84.1 | 88.6 | 88.5 | 83.8, 94.0 | 87.8* | 88.4, 87.0 | 93.0 | 88.4, 97.0 |

*Denotes significant higher balanced accuracy than ConvNet by DeLong's test ($p < 0.05$).

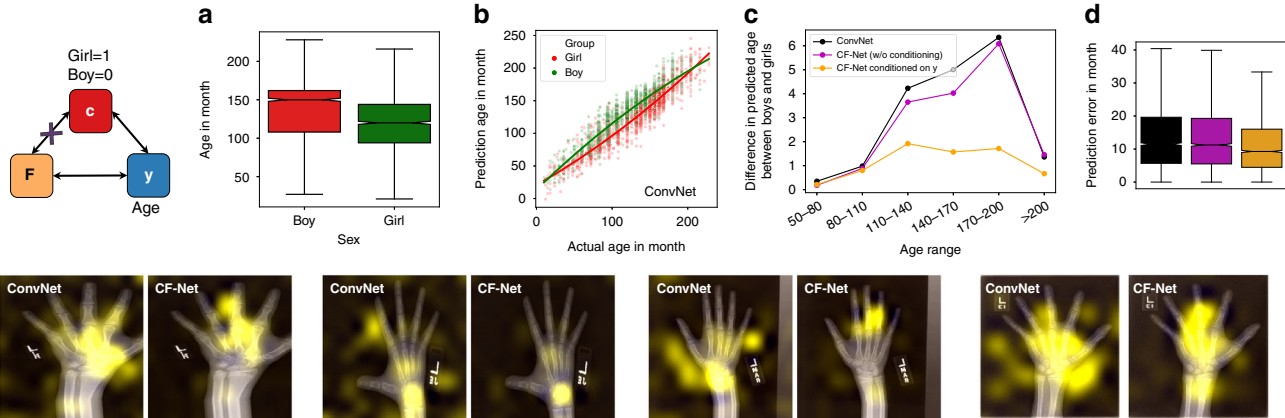

**Fig. 5 Bone-age prediction from hand X-ray images. a** Difference in the age distribution between $n = 6,833$ boys and $n = 5,778$ girls of the RSNA bone-age dataset ($p < 0.0001$, two-tailed two-sample $t$-test). **b** Ground truth vs. predicted age of the ConvNet. ConvNet tended to predict higher age for girls than boys, indicating a confounding effect of sex. **c** This prediction gap between boys and girls was more pronounced in the age range of 110–200 months, but was significantly reduced by CF-Net, which modeled the dependency between **F** and **c** on a **y**-conditioned cohort. **d** Absolute prediction error (in months) of $n = 3,153$ testing subjects produced by ConvNet and CF-Net with (or without) conditioning. Boxplots are characterized by minimum, first quartile, median, third quartile, and maximum. CF-Net with conditioning resulted in the most accurate prediction ($p < 0.0001$, two-tailed two-sample $t$-test).

distributions between the two genders (see "Methods" section). By doing so, CF-Net successfully reduced the sex-related gap in age prediction (Fig. 5c, Supplementary Figs. 11–13). Moreover, the prediction accuracy of CF-Net with **y**-conditioning was

significantly higher (absolute error $11.2 \pm 8.7$ months) than that of the baseline ConvNet and CF-Net without **y**-conditioning (two-tailed $p < 0.0001$, one-sample $t_{3152} = 14.2$, Fig. 5d). The saliency maps of CF-Net were more localized on anatomical

structures than those of ConvNet, indicating that the widespread pattern leveraged by ConvNet might be redundant and relate to confounder-related cues. Note, as in the prior experiment, the accuracy of CF-Net was similar to ConvNet when training $\mathbb{CP}$ on all subjects available (without conditioning on **y**).

## Discussion

Accurate modeling of confounders is an essential aspect in analyzing medical image[2]. For example, traditional machine-learning models rely on precomputed features from which confounding effects are regressed out a priori[7,10,6]. This topic, however, is largely overlooked by deep-learning applications as researchers shift attention to designing deeper and more powerful network architectures to achieve higher classification/regression accuracy[42–44]. Indeed, end-to-end learning of deep models often is superior to traditional machine-learning methods relying on precomputed features. For example, the ConvNet baseline reported a higher accuracy (BAcc: 71.6%) in the HIV experiment than applying a traditional SVM classifier to the 298 brain regional measurements extracted by FreeSurfer[45] (BAcc: 69.5%). The more accurate predictions of such deep models are in part due to increased sensitivity to subtle group differences, which also heightens the risk of biasing findings as these subtle differences may relate to confounders. For example, on the NCANDA dataset, ConvNet produced the highest prediction accuracy on the entire cohort, which was partially attributed to the confounding effect of PDS. Therefore, the superiority of a prediction model for medical imaging applications should be defined with respect to its predictive power and impartiality to confounders. However, the a priori strategies (used by traditional machine learning) for training impartial predictors do not work for end-to-end learning models as learning is based on extracting features on-the-fly from raw images. While recent advances in adversarial learning have shed light on this problem, existing deep models were only designed to tackle specific confounding effects such as scanner difference or dataset harmonization[46–48]. Here, we propose a deep-learning architecture for systematically and accurately modeling confounders in medial image applications based on adversarially training a confounding predictor $\mathbb{CP}$ (see Fig. 2). $\mathbb{CP}$ can be used to remove confounding effects of any layer of a generic deep model, such as the entire feature extractor in the MRI experiments or a submodule of the extractor in the bone-age experiment.

By explicitly modeling the confounding effect in the feature-learning process, CF-Net bypasses the need of matching cohorts with respect to confounders, which generally reduces the sample size and thus negatively impacts generalizability of the model[13]. However, training models on confounded data now requires evaluating the fairness of model predictions with respect to confounders. In line with the concept of group fairness or demographic parity[49,50], one can do so by examining whether the predictive power of the model varies across different validation subsets. We did so by measuring the difference between the testing accuracy recorded on the whole (confounded) cohort and on the **c**-independent (unconfounded) subset. We viewed this difference as a metric for the severity of the confounding effects: the larger the difference, the more confounded the model. Another way of defining validation subsets is to group testing subjects according to their confounder values (see Figs. 3b, c and 4b, c). In all three experiments, CF-Net achieved more balanced prediction accuracies across those subsets than ConvNet, further highlighting the fairness of the CF-Net model.

Another important property of CF-Net is its ability to model continuous confounders (e.g., age), whereas most existing fair machine-learning methods[17,23,51–53,25] are confined to binary or

discrete confounders (e.g., gender). This improvement is achieved by our loss function based on squared correlation (see "Methods" section), which encourages statistical mean independence between the derived high-dimensional features and a scalar extraneous variable (in our case, a confounder). When applying this adversarial loss to subjects from the **y**-conditioned cohort, CF-Net outperformed other state-of-the-art deep models in classification accuracy. Although this improvement did not meet the significance level after multiple-comparison correction, CF-Net resulted in impartial features and unbiased model interpretation according to the experiments in Supplementary Information. These complementary tests thoroughly assessed the confounding effects in the underlying feature space and extended beyond the aforementioned fairness evaluation defined on prediction outcomes.

Learning confounder-free features is particularly challenging when the confounder inherently correlates with the prediction labels[53,25], such as in the three experiments presented here. As pointed out in refs. [53,25], general fair/invariant feature-learning frameworks could potentially be harmful in this situation as it is impossible to derive features that are simultaneously discriminative with respect to **y** and independent with respect to **c**. This argument was supported by our experiments (see also Supplementary Figs. 1, 12, and 13) showing comprised prediction accuracy or increased bias in the learned features when training $\mathbb{CP}$ on all subjects in the training set. To address this issue, we proposed here to learn the direct link between **F** and **c** by modeling their conditional independence in a **y**-conditioned cohort, i.e., subjects with **y** values confined to a certain range. The practice of conditioning has been a standard approach in the statistics literature for studying the relation between two variables while controlling the mediation from a third variable[54,55]. In the case of binary classification, conditioning on **y** is equivalent to fixing **y** to either group so that the inherent correlation between **c** and **y** is removed. However, the specific group chosen to model the conditional dependency is application-specific. In the HIV experiment, the relation between **F** and **c** was supposed to capture normal aging, which could only be studied on the control group (fixing **y** = 0) as HIV accelerates brain aging[26–28]. In the NCANDA experiment, boys (**y** = 0) or girls (**y** = 1) would have been theoretically suitable to train $\mathbb{CP}$ being impartial to PDS. Of the two cohorts, training conditioned on boys resulted in more impartial predictions as this cohort covered the full range of PDS values, while lower PDS scores were not well represented in the girl-conditioned cohort as adolescent girls are generally more mature than boys of the same age. When predicting a continuous variable, we proposed to define the **y**-conditioned cohort by selecting samples whose **y** was confined to a fixed interval and decorrelating **y** and **c** via bootstrapping. In the bone-age experiment, the interval was selected as the full width at half maximum (FWHM)[56] of the overall age distribution, which approximately encompassed 80% of the training subjects and focused only on the age range with sufficient samples (Supplementary Fig. 10). This well-represented age interval facilitated the decorrelation with respect to gender and resulted in a large **y**-conditioned cohort for training $\mathbb{CP}$. Another strategy for defining the interval (not explored in this paper) is to model the interval as a hyper-parameter, whose optimal setting is determined via parameter exploration during nested cross-validation. Alternatively, one can bypass the need of selecting the interval by using data-driven matching procedures (e.g., a bipartite graph matching[57] or greedy algorithm[7]), which in our experiments produced similar accuracy scores as the one based on the FWHM criteria and bootstrapping.

Based on these different **y**-conditioning strategies, medical researchers can use CF-Net to train deep models on cohorts not strictly matched with respect to confounders without discarding

unmatched samples. However, this does not mean that there is no need to keep the confounders in mind when recruiting participants for medical imaging studies. For all learning models, performing analysis on confounder-matched cohorts with sufficient samples remains a fundamental strategy to disentangle biomarkers of interest from the effects of confounders. For example, in the bone-age experiment, recruiting enough age–gender-matched samples resulted in a large **y**-conditioned cohort that reduces the risk of overfitting during the training of $\mathbb{CP}$. Conversely, if two cohorts have completely different distributions with respect to a confounder (e.g., one has participants with strictly larger age than the other), there is no guarantee that any method, including ours, can remove the bias in a purely data-driven fashion. Therefore, in the study-design stage, defining potential confounders for a specific medical application may require domain-specific knowledge to maximize the power of CF-Net in practice.

A limitation of our experiments was the focus on single confounders that were known a priori. To model unknown confounders, we aim to explore coupling CF-Net with causal discovery algorithms (such as refs. [58–60]). In case predictions are biased by multiple confounders, we would need to extend $\mathbb{CP}$ to predict multiple outputs (one for each confounder) or add for each confounder a $\mathbb{CP}$ component to CF-Net. In the simple scenario that the confounding variables are conditionally independent with respect to **y**, each $\mathbb{CP}$ component can be trained on a separate **y**-conditioned cohort uniquely defined for each confounder. However, theoretical and practical ways in modeling high-order interactions between confounders require further investigation.

While we were able to visualize the HIV and sex effect by computing saliency maps[61] inferred from the predictor $\mathbb{P}$, the same technique is not directly applicable to visualize confounding effects from $\mathbb{CP}$ due to the adversarial training. An alternative could be deriving saliency maps from $\mathbb{CP}$ retrained on the features learned by the baseline ConvNet (e.g., Supplementary Fig. 2), i.e., a model that substantially captures the confounding effect.

Finally, we abstained from determining the optimal implementation of the proposed confounder-free modeling strategy by performing extensive exploration of network architectures. Instead, we relied on some of the most fundamental network components used in deep learning. This rather basic implementation still recorded reasonable prediction accuracies, so the findings discussed here are likely to generalize to more advanced network architectures.

## Methods

**Materials**. This study used multiple medical imaging data sets to evaluate different aspects of our proposed confounder-free neural network, described briefly herein. In addition, experiments on synthetic data sets are included in Supplementary Fig. 1, which shows the efficacy of our proposed framework in controlled settings.

*HIV dataset*: Our first task aimed at predicting the diagnosis of HIV patients vs. control subjects[62]. Participants ranged in age between 18 and 86 years and were all scanned with a T1-weighted MRI. All study participants provided written informed consent, and the study was approved by Institutional Review Board (IRB) at Stanford University (Protocol ID: IRB-9861) and SRI International (Protocol ID: Pro00039132). HIV subjects were seropositive for the HIV infection with CD4 count >100 $\frac{\text{cells}}{\mu\text{L}}$ (average: 303.0). Construction of the **c**-independent subset was based on the matching algorithm[7] that extracted the maximum number of subjects from each group in such a way that they were equal in size and identically distributed with respect to the confounder values. For each HIV subject, we selected a control subject with minimal age difference and repeated this procedure until all HIV subjects were matched or the two-tailed $p$ value of the two-sample $t$-test between the two age distributions dropped to 0.5. The MR images were first preprocessed[7] by denoising, bias-field correction, skull striping, and affine registration to the SRI24 template[63]. The registered images were then downsampled to a $64 \times 64 \times 64$ volume[64] based on spline interpolation to reduce the potential overfitting during training and to enable a large batch size. Prediction

accuracy of the deep models was determined via fivefold cross-validation. For each training run, MRIs were augmented to provide sufficient number of samples for the model to be trained on. As in ref. [65], data augmentation produced new synthetic 3D images by randomly shifting each MRI within one voxel and rotating within 1° along the three axes. The augmented dataset included a balanced set of 1024 MRIs for each group (control and HIV). Assuming that HIV affects the brain bilaterally[7,66], the left hemisphere was flipped to create a second right hemisphere. During testing, the right and flipped left hemispheres of the raw test images were given to the trained model, and the prediction score averaged across both hemispheres was used to predict the individual's diagnosis group. Last, a saliency map was computed[61] for the right hemisphere of each test image quantifying the importance of each voxel to the final prediction.

*NCANDA dataset*: Experiments were performed on the baseline T1 MR images of 334 boys and 340 girls from the NCANDA study (Public Release: NCANDA_PUBLIC_BASE_STRUCTURAL_V01[67]). Adult participants and the parents of minor participants provided written informed consent before participation in the study. Minor participants provided assent before participation. The IRB of each site approved the standardized data collection and use[39]. All subjects met the no-to-low alcohol drinking criteria of the study, and there was no significant age difference between boys and girls ($p > 0.5$, two-sample $t$-test). Pubertal stage was determined by the self-assessment pubertal development scale (PDS). Procedures for preprocessing, downsampling, and classifying the MRI were conducted according to the HIV experiment.

*Bone-aging dataset*: The RSNA Pediatric Bone Age Machine Learning Challenge was based on a dataset consisting of 14,236 hand radiographs (12,611 training sets, 1425 validation sets, and 200 test sets)[40]. We experimented on the 12,611 training images with ground-truth bone age ($127.3 \pm 41.2$) and the ConvNet model publicly released on the *Kaggle* challenge page[41]. In total, 3914 boys and 3518 girls, or 80% of the training subjects (Fig. 5a), had bone ages between 75 months and 175 months (the FWHM of the age distribution, Supplementary Fig. 10). Confined to this age range, we used bootstrapping[68] to generate 1000 boys and 1000 girls within each 10-month interval. This procedure resulted in a **y**-conditioned cohort of 10,000 boys and 10,000 girls strictly matched with respect to bone age ($p = 0.19$, two-tailed two-sample $t$-test).

**Confounder-free neural network (CF-Net)**. Suppose we have $N$-training MR images $\mathcal{X} = \{\mathbf{X}_i\}_{i=1}^N$ and their corresponding target-prediction values $\mathbf{y} = \{y_i\}_{i=1}^N$, where $y_i \in [0, 1]$ for classification problems and is a continuous variable for regression problems. Let us assume that the study is confounded by a set of $k$ variables and their values are denoted by $\mathbf{C} = \{\mathbf{c}_i\}_{i=1}^N$, where each $\mathbf{c}_i = [c_i^1, ..., c_i^k]$ is a $k$-dimensional vector denoting the $k$ confounders of subject $i$. To train a deep neural network for predicting the target value for each input MR image $\mathbf{X}$, we first apply a Feature Extraction ($\mathbb{FE}$) network to the image, resulting in a feature vector $\mathbf{F}$. A Classifier ($\mathbb{P}$) is built on this feature vector to predict the target $\mathbf{y}$ for the input $\mathbf{X}$. This ensures the discriminative power of the learned features and defines the baseline architecture of ConvNet. Now, to guarantee that these features are not biased to the confounders, we propose our end-to-end architecture as in Fig. 2. Specifically, we build another network (denoted by $\mathbb{CP}$) for predicting the confounding variables from $\mathbf{F}$ and backpropagate this loss to the feature-extraction module in an adversarial way. We train $\mathbb{CP}$ only on a **y**-conditioned cohort consisting of subjects whose target **y** values are uncorrelated with all $k$ confounders. We define the **y**-conditioned cohort as $\mathcal{X}_\rho$ with $\rho_i = 1$ if $\mathbf{X}_i \in \mathcal{X}_\rho$, and $\rho_i = 0$ otherwise. The confounders associated with the **y**-conditioned cohort are correspondingly denoted as $\mathbf{C}_\rho$. As a result, the feature extractor learns features that minimize the **y** predictor loss while being conditionally independent of the confounder by maximizing the loss of $\mathbb{CP}$ for $\mathcal{X}_\rho$.

Each of the above networks have some underlying trainable parameters, defined as $\boldsymbol{\theta}_{fe}$ for $\mathbb{FE}$, $\boldsymbol{\theta}_p$ for $\mathbb{P}$, and $\boldsymbol{\theta}_{cp}$ for $\mathbb{CP}$. $\mathbb{P}$ forces the feature extractor to learn features to better predict $y_i$ by backpropagating the prediction loss. Let $\hat{y}_i = \mathbb{P}(\mathbb{FE}(\mathbf{X}_i; \boldsymbol{\theta}_{fe}); \boldsymbol{\theta}_p)$ be the predicted $y_i$, then the prediction loss can be characterized by binary cross-entropy $l(y_i, \hat{y}_i) = -y_i \log \hat{y}_i - (1 - y_i) \log (1 - \hat{y}_i)$ for classification and by the mean-squared error $l(y_i, \hat{y}_i) = (y_i - \hat{y}_i)^2$ for regression. Finally, the prediction loss for the entire cohort is

$$L_p(\mathcal{X}, \mathbf{y}; \boldsymbol{\theta}_{fe}, \boldsymbol{\theta}_p) = \frac{1}{N} \sum_{i=1}^N l(y_i, \hat{y}_i). \tag{1}$$

Similarly, with $\hat{c}_i = \mathbb{CP}(\mathbb{FE}(\mathbf{X}_i; \boldsymbol{\theta}_{fe}); \boldsymbol{\theta}_{cp})$, we define the surrogate loss of confounder prediction for the **y**-conditioned cohort as

$$L_{cp}(\mathcal{X}_\rho, \mathbf{C}_\rho; \boldsymbol{\theta}_{fe}, \boldsymbol{\theta}_{cp}) = -\sum_{\kappa=1}^k \text{corr}^2(\mathbf{c}^\kappa, \hat{\mathbf{c}}^\kappa), \tag{2}$$

where $\text{corr}^2(.,.)$ is the squared correlation between its inputs and $\mathbf{c}^\kappa$ defines the vector of $\kappa$th confounding variable in $\mathbf{C}_\rho$. Hence, the overall objective of the network with a trade-off hyperparameter $\lambda$ is

$$\min_{\boldsymbol{\theta}_{fe}, \boldsymbol{\theta}_p} \max_{\boldsymbol{\theta}_{cp}} L_p(\mathcal{X}, \mathbf{y}; \boldsymbol{\theta}_{fe}, \boldsymbol{\theta}_p) - \lambda L_{cp}(\mathcal{X}_\rho, \mathbf{C}_\rho; \boldsymbol{\theta}_{fe}, \boldsymbol{\theta}_{cp}). \tag{3}$$

This scheme is similar to the GAN formulations[29] with a min–max game between two networks. In our case, $\mathbb{FE}$ extracts features that minimize the classification criterion, while fooling $\mathbb{CP}$ (i.e., making $\mathbb{CP}$ incapable of predicting the confounding variables). Hence, the saddle point for this optimization objective is obtained when the parameters $\theta_{fe}$ minimize the classification loss while maximizing the loss of the confounder-prediction module. Simultaneously, $\theta_p$ and $\theta_{cp}$ minimize their respective network losses.

**Implementation**. After normalizing confounder values to z scores, we optimize Eq. (3) based on the practice used in GANs. In each iteration, we first train $L_p$ on a mini batch sampled from all available training data. The loss of $L_p$ was backpropagated to update $\theta_{fe}$ and $\theta_p$. With $\theta_{fe}$ fixed, we then minimize $L_{cp}$ to update $\theta_{cp}$ by computing the correlation of Eq. (2) over subjects of a mini batch sampled from the **y**-conditioned cohort. Finally, with $\theta_{cp}$ fixed, $L_{cp}$ is maximized by updating $\theta_{fe}$ with respect to the correlation loss defined on a mini batch from the **y**-conditioned cohort.

With respect to the network architecture used in the experiments, we followed the design of $\mathbb{FE}$ in refs. [69,64] that contained 4 stacks of $2 \times 2 \times 2$ 3D convolution/ReLu/batch-normalization/max-pooling layers, yielding 4096 intermediate features. Each of $\mathbb{P}$ and $\mathbb{CP}$ was a two-layer fully connected network. We set $\lambda$ to 1 (see Supplementary Fig. 5) and used a batch size of 64 subjects and Adam optimizer with a learning rate of 0.0002. For the 2D X-ray experiment, the $\mathbb{FE}$ and $\mathbb{P}$ components complied with the feature extractor and predictor defined in ref. [41].

**Reporting summary**. Further information on research design is available in the Nature Research Reporting Summary linked to this article.

## Data availability

All data used in this paper are described and their respective references are cited in the "Materials" subsection of the Methods section. For the HIV dataset, as previously described[69], patients were recruited by referral from local outpatient HIV/AIDS-treatment centers, presentations by project staff, and distribution of flyers at community events. Control participants were recruited by referral from patient participants, Internet posting, flyers, and word of mouth. This dataset is not accessible by the public. The NCANDA data used here are from the data release NCANDA_PUBLIC_BASE_STRUCTURAL_V01 (digital object identifier 10.7303/syn11541569)[67] distributed to the public according to the NCANDA Data Distribution agreement https://www.niaaa.nih.gov/research/major-initiatives/national-consortium-alcohol-and-neurodevelopment-adolescence/ncanda-data. Recipient acknowledges that the collection of NCANDA data was approved by the IRB of the local collection sites in accordance with the Department of Health and Human Services regulations at 45 CFR Part 46. The Bone Age dataset is publicly available at https://www.kaggle.com/kmader/rsna-bone-age. It is released by the RSNA RIC as a machine-learning challenge for predicting pediatric bone age[40].

## Code availability

Custom scripts, including those for generating the synthetic dataset, have been made available at https://github.com/qingyuzhao/br-net/(https://doi.org/10.5281/zenodo.4122448). Additional preprocessing scripts may be accessed upon request.

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

## Acknowledgements

This work was made possible with support by National Institutes of Health (NIH) Grants AA021697, AA021695, AA021692, AA021696, AA021681, AA021690, AA021691, AG066515, and MH113406. The content is solely the responsibility of the authors and does not necessarily represent the official views of the NIH. This study also benefited from Stanford University School of Medicine Department of Psychiatry & Behavioral Sciences 2021 Innovator Grant Program as well as Stanford Institute for Human-centered Artificial Intelligence (HAI) AWS Cloud Credit.

## Author contributions

All authors (Q.Z., E.A., and K.M.P.) contributed to experimental design, data analysis, and paper writing. The authors declare no competing interests.

## Competing interests

The authors declare no competing intersts.
