## [Peer Review File · Nature Communications]

Reviewers' Comments:

Reviewer #1:

Remarks to the Author:

<Summary of review>

This paper proposes confounder-free neural networks, especially considering medical applications. In my understanding, this paper holds two main claims:

- (1) The standard CNN trained over medical images is substantially influenced by con-founder.
- (2) The issues can be alleviated by adversarial training over the y-conditioning cohort.

While the experimental results seem extensive and proposed methods have some methodological novelty, several technical details need to be clarified.

<Major comments / questions>

- (1) Technical detail of the proposed method.

In my understanding, the main technical advance in this paper is the introduction of y-conditioning technique (while using correlation measurements is also somewhat new), which helps adversarial training to focus on removal of direct influences between c and F and ignoring the relationships between c and y

Q1-1. When and how does y-conditioning apply? Does it applied once before training, or do we select a y-conditioned subset for each mini-batch during training? Besides, when the c is continuous multivariate variable as assumed in 4.2, exact y-conditioning is difficult. How exactly conduct y-conditioning? How does the performance on y-conditioning affect the performance?

Q1-2. Why do you choose the correlation scores instead of the log-likelihood, which is commonly used in prior works, e.g., [4, 54]? How does the performance changes if we change the optimization metrics?

- (2) Experimental design.

From my viewpoints, the main strength of the paper is extensive experimental results on various medical image datasets. However, many experimental design are unclear for me. Please clarify below.

Q2-1. Why do you compare the performance on (a) c-dependent cohorts and (b) c-independent cohorts?

For example, authors claim that

"While the accuracy, prediction and recall of CF-Net were similar to therst experiment, the balanced accuracy of 3D ConvNet reduced to 68.4%, which now was significantly lower ($p = 0:035$, DeLong's test) than the BAcc of CF-Net (74.2%)" in page 4.

However, the performance degradation is natural since the number of training datasets is decreased. So, I don't think that the performance drop directly indicates the CNN is substantially biased by confounder.

Q2-2. Related to the Q2-1, I do not fully understand what metric the paper try to maximize/minimize, and what application scenario they are considering. In other words, it is not clear to me why the authors chose the specific metrics reported in this paper. As discussed in the

fair representation learning articles, which tackle related technical problems, there are many possible metrics to compare the performance. As the different metrics measure the different aspects, I strongly recommend adding a discussion about why we need to focus on the metrics reported in this paper.

Ref: "21 fairness definitions and their politics",
<https://fairmlbook.org/tutorial2.html>

Related to the above comments, some statements in the introduction needs to be clarified. For example, "These different methods, however, fail to provide the means to understand the potential effect of the confounder on the outcome" => why does it a problem?

Q2-3. While the author only compares vanilla CNN and the proposed method, [54] can be applied for continuous variable settings. Why don't you compare it besides?

Q2-4. How does the weighing parameter λ is selected? How the choice of hyperparameter affect the results? The article describes that they used 5-fold cross-validation thorough out the paper, but never mentioned the validation procedures.

<Minor comments>

- Reference on page 2 contains "?".
- Some parts of section 2 discuss the balance between precision and recall. However, the balance can be controlled by changing the thresholds, and therefore I'm not sure what can be said from the balance itself. Instead, PR-curve is a better metric in the case.
- Why the performance on the c-independent cohort does not drop for the proposed method, even though the number of total examples is decreased?
- While the authors claim that "CF-Net could alleviate the confounding effect in the prediction according to the higher accuracy measured on a 'c-independent' sub-cohort compared to the ConvNet" (on the top of page 7), I do not think we can make this conclusion only from this results, as there is no confounding effect when we use the c-independent sub-cohort. Please clarify the statements.
- p9, line 280 needs a comma after c_{ρ} .
- While the equations do not contain expectations over the batch, I think the proposed method also uses mini-batch training. Please correct the equations.
- Section 4.3 should contain optimization detail for reproduction.

Reviewer #2:

Remarks to the Author:

Zhao and colleagues present a very interesting manuscript on the important topic of statistical confounders in deep learning models in medical imaging. They offer a novel adversarial approach to remove confounding effects from end-to-end classification and regression models by directly modelling the link between the confounding and the input features in sub-samples of the data where there is no dependence between the confounder and the outcome of interest.

The study has a strong rationale and is very well-considered in the design of the confounder-free network, which I found compelling. They have cleverly updated classical statistical principles for the context of deep learning and supported their ideas with three separate experiments, which is a real strength. The application of the CF-Net to continuous variables is particularly interesting.

However, when it comes to the reporting of the Results, I began to get a rather confused and I couldn't properly evaluate just how effective their new approach is relative to the ConvNet baseline. There is a lack of consistency between the three experiments in terms of exactly what results are reported. They don't include the whole-cohort result from the NCANDA experiment in Table 2, and there isn't even a results table for the bone-age experiment. For example, I think it would be important to know whether the differences in MAE reported in Figure 5d are significant.

Also, the t-SNE and distance correlation results are only reported for the HIV experiment. To me, it makes sense that all three experiments are reported in a consistent way, unless there's a clear rationale for omitting some metrics in certain cases.

More fundamentally, I'm not sure I understood the logic of the Results. My initial assumption was that the CF-Net was designed to be beneficial (i.e., should outperform the ConvNet) when using confounded data, and that the performance would be equivocal when using the confound-independent data (in other words, when there is no confound, the CF-Net isn't necessary). However, the authors' logic seems to be the opposite of that, as in the Discussion (line #189) they say "CF-Net could alleviate the confounding effect in the prediction according to the higher accuracy measured on a 'c-independent' sub-cohort compared to the ConvNet." Perhaps I'm getting the wrong end of the stick here. Is the idea that the performance in the whole cohort and confound-independent cohorts should be the same if confounds are properly accounted for? But if not controlled for (i.e., ConvNet results), the performance in whole cohort is artificially high because of the confound? I think it is essential this should be clarified (both to me and in the text), as the key results hinge on this point.

Results. In the confounded HIV dataset (i.e., the whole cohort), presumably the DeLong's test was not significant? This should be reported as it suggested that CF-Net did not outperform the 3D ConvNet in this instance. Perhaps add 4 DeLong test results to Table 1 so that it's clear when there's a difference between the CF-Net and the 3D ConvNet. Would it be worthwhile using the DeLong test for all three experiments in fact?

In the Abstract, it says that the authors' method results "in superior prediction accuracy compared to the baseline and recent invariant feature learning frameworks". The results of the comparison with recent invariant feature learning frameworks seems to only appear in the Supplementary material and are not mentioned in the main text, unless I missed it. I recommend that anything that is mentioned in the Abstract should also appear (even if only briefly) in the main text. Also, on scrutiny of Table S1, I'm not convinced that the CF-Net significantly outperforms the methods by Zafar et al., or Sadeghi et al., in HIV classification, so this claim requires some further information to back it up.

Figure 3 f and g. The authors interpret these patterns of saliency and as being age-related and HIV-related respectively. I'm not sure that I agree with this interpretation as both age and HIV affect brain structure in a more distributed way than is apparent in the figure. Furthermore, the regions mentioned in the Results section (e.g., cerebellum) are not visible at in the figure. To make a more convincing argument that the saliency maps for the different networks do indeed resemble ageing or HIV I think at the very least they need to include several more brain slices. To go further, it would be straightforward to conduct a VBM analysis of group age and of HIV group using the same data, then see whether the voxels where volumetric differences are associated with these factors are similar to the saliency maps.

In the Discussion they mention that they ran an SVM on FreeSurfer features on the HIV dataset and achieved 69.5%. They say that this is significantly lower than the deep models, but it's not that much lower than the ConvNet (accuracy = 71.6%) and they can only really say 'significant' if they tested that difference explicitly (e.g., DeLong's test). Moreover, I contend that this is not really a fair comparison. FreeSurfer ROIs are by design reductive, and it's quite possible to use whole brain voxelwise data (e.g., VBM maps) as input in an SVM. My guess is that this would achieve better performance as more of the original signal has been retained. I realise that the authors' statement in the Discussion is something of an aside, but it really should be backed up by more substantive evidence or removed.

Methods – the MRI volumes for the HIV dataset were re-scaled to 64^3 . This will naturally reduce the amount of information contained in each image. How did the authors arrive at 64^3 as the right input, and what sort of interpolation was done? Presumably they did the same re-scaling to the NCANDA data?

Figure 5c – I'm not clear what exactly has been plotted here. Can the authors please clarify? Generally, I suggest not to use bar plots for anything other than count data and I think the y-axis is continuous here.

Did the authors consider visualising the confounding effects from the adversarial component of the network (i.e., CP)?

To what extent does the data augmentation scheme (especially the left-right flipping) influence the saliency maps?

The precise construction of the Feature Extraction (FE) network is likely to have a big impact on exactly what features get used in the model. For bone age they used VGG, but for the MRI experiments the FE network was trained from scratch. How did they decide on the specific configuration on the network?

On a more philosophical note, I think some discussion of the following would be informative for readers. If this network can remove the influence of confounders, does this mean that there is less need to match samples for confounders (e.g., age) when recruiting? Or are we not at that stage yet?

Reviewer #3:

Remarks to the Author:

General Comments:

In the manuscript "Training Confounder-Free Deep Learning Models for Medical Applications", Zhao et al. propose a deep learning approach to learn features that are invariant to pre-specified confounding factors. The authors evaluated the method on three problems, diagnosing of HIV from MRIs, identifying morphological sex differences in adolescence, and determining the bone age from X-ray images of children. The authors demonstrated that the proposed approach shows robust while doing subgroup analysis on the confounding factors. Suggesting the potential for the trained model to perform the desired tasks using less information from the confounding factors.

Major comments:

1. The authors proposed Confounder-Free Neural Network with a GAN-like formulation of a min-max game between the classification prediction network and the confounder prediction network. The formulation seems straightforward. It will be helpful if the authors can draw connections and highlight the differences between the proposed method and the existing methods in the literature.
2. Is the proposed method CF-Net the same as the BR-Net in the authors' authors' prior publication titled "Representation learning with statistical independence to mitigate bias"? If so, this work's contribution might be at the lighter end as the novel part of this manuscript is applying the previously proposed methods on three medical image applications. If not, it will be helpful to make this clear in the paper and highlight the distinctions.
3. Since this paper's main contribution is the proposal of the CF-Net method, further discussion on the design choices is needed, especially the choices of $L_{\{cp\}}$ and the training of CP only on a y -conditioned cohort. These choices should be justified and compared with alternative methods. For example, why only train on a single y -conditioned cohort instead of having some training scheme that utilizes all possible y -conditioned cohorts. And why using squared correlation as the loss function instead of other alternatives.
4. Please describe how the operating points were selected for results in Tables 1 and 2.
5. Please describe the approach to select the confounder-independent cohort.
6. Please provide confidence intervals for the results in Tables 1 and 2.
7. L89: "Only 36.3% of the young HIV subjects were correctly labeled." With this context, how should we understand the evaluation metrics in Table 1 given these incorrect labels?
8. Section 2.2: Why are there asymmetric results such that conditioning on boys has different results from conditioning on girls?
9. Discussion: it will be great if the authors can comment on how this method can be extended to scenarios without knowing the confounding variables of interest.
10. I don't quite understand the training approach of only training CP on a cohort of the same y . Given a binary dataset with balanced $y=0$ and $y=1$, this is essentially only using half of the dataset to train CP. Why can't we train with both?
11. For this GAN-like min-max training, it will be informative for providing the loss functions with

breakdowns of L_p and L_{cp} over the train and tune set in the Supplementary. I'm also curious of how CP performs in predicting confounding variables after the model converges, this will highlight to what extent can the confounding information be removed in F.

Minor comments:

1. L95: upper left region
2. Section 2.1: If I understand correctly, the proposed CF-Net is compared against the same architecture but without the CP part. It might be easier to make this clear and rename the 3D ConvNet as ConvNet. 3D isn't the focus, and if you need to add 3D to the name, you might also want to update CF-Net to 3D CF-Net.
3. L160: Can you elaborate a bit more on why a more localized saliency map is preferred in predicting age from X-ray?

Submission ID: NCOMMS-20-23428

Training Confounder-Free Deep Learning Models for Medical Applications

We appreciate the valuable comments and feedback from the reviewers. To address the comments while keeping the focus of the article, we explained in further detail the main novelty of the proposed approach (i.e., ‘ \mathbf{y} -conditioning’) and moved other relevant analysis to the supplement. Point-to-point responses to the reviewers’ comments and the modifications in the manuscript are listed in the following. We also provide a version of the revised manuscript, in which all the modified text are typeset in blue.

Response to Comments from Reviewer #1:

We thank the reviewer for the positive feedback on the idea of our paper. We carefully addressed the comments. Please see the responses below

Q1-1a. How exactly conduct \mathbf{y} -conditioning?

Response: The ‘ \mathbf{y} -conditioning’ refers to training $\mathbb{C}\mathbb{P}$ on a ‘ \mathbf{y} -conditioned’ cohort, i.e., subjects of the training data with \mathbf{y} values being the same (e.g., controls) or confined to an interval (e.g., age range). This revision clarified this point in Section Introduction

“We therefore specifically train $\mathbb{C}\mathbb{P}$ on a ‘ \mathbf{y} -conditioned’ cohort, i.e., samples of the training data whose \mathbf{y} values are confined to a specific range”

In the HIV experiment, we conducted ‘ \mathbf{y} -conditioning’ by

“On the four folds used for training, we used data augmentation to generate two cohorts of equal size and confined the predictions of age by $\mathbb{C}\mathbb{P}$ to controls (i.e., the \mathbf{y} -conditioned cohort was defined by $\mathbf{y} = 0$).”

In the NCANDA experiment, we stated

“the modelling of the PDS effect could be conditioned on $\mathbf{y} = 0$ (boys) or $\mathbf{y} = 1$ (girls)”

In the bone-age experiment, we clarified

“the \mathbf{y} -conditioned cohort could not be defined with respect to a fixed prediction outcome. Instead, we applied the $\mathbb{C}\mathbb{P}$ component to a matched dataset, where the 3,914 boys had the same age range as the 3,518 girls (i.e., $y \in [75 \text{ months}, 175 \text{ months}]$)”

Q1-1a. When and how does \mathbf{y} -conditioning apply? Does it applied once before training, or do we select a \mathbf{y} -conditioned subset for each mini-batch during training?

Response: In line with the prior response, we simply defined the \mathbf{y} -conditioned cohort once from all the training data. During training, we sampled mini-batches from the predefined \mathbf{y} -conditioned cohort. Note, while we did not perform \mathbf{y} -conditioning for each mini-batch, these

two approaches result in similar models as the mini-batch gradient descent is simply a numerical algorithm that optimizes the objective function defined on the whole cohort. We clarified this point in Section 4.3

“With θ_{fe} fixed, we then minimize L_{cp} to update θ_{cp} by computing the correlation of Eq. (2) over subjects of a mini-batch sampled from the \mathbf{y} -conditioned cohort. Finally, with θ_{cp} fixed, L_{cp} is maximized by updating θ_{fe} with respect to the correlation loss defined on a mini-batch from the \mathbf{y} -conditioned cohort.”

How does the performance on \mathbf{y} -conditioning affect the performance?

Response: In the revision, we further clarified the impact of \mathbf{y} -conditioning on prediction accuracy with respect to the NCANDA study (see Table 2, Fig. 4) and bone-age study (Fig. 5). In the NCANDA experiment (Section 2.2), we emphasize that CF-Net conditioned on boys was superior than CF-Net without conditioning

“Confining the training of $\mathbb{C}\mathbb{P}$ to boys resulted in more accurate predictions (see Fig. 4(c-e)). In doing so, CF-Net recorded the highest balanced accuracy on the \mathbf{c} -independent subset, had the smallest gap in accuracy across subjects at different pubertal stages, was significantly more accurate in prediction at the early pubertal stage (two-tailed $p = 0.039$ DeLong’s test), and produced features significantly less predictive of PDS ($p < 0.001$ one-sample $t_{333} = 12.2$, Supplement Fig. S7, S8) than ConvNet.”

In the bone-age study (Section 2.2), we clarified that

“the prediction accuracy of CF-Net with \mathbf{y} -conditioning was significantly higher (absolute error 11.2 ± 8.7 months) than that of the baseline ConvNet and CF-Net without \mathbf{y} -conditioning (two-tailed $p < 0.001$, one-sample $t_{3152} = 14.2$, two-sample t -test, Fig. 5(d)).”

Q1-1b. Besides, when the \mathbf{c} is continuous multivariate variable as assumed in 4.2, exact \mathbf{y} -conditioning is difficult.

Response: This comment by the reviewer points out a weakness in our original formulation for the case that \mathbf{y} is continuous and correlates with multiple \mathbf{c} (confounders) regardless of \mathbf{c} being continuous or discrete. In the revision, we improved the notation in Section 4.2, kept the focus of each experiment on a single confounder, and discussed this potential weakness in the new limitation section.

We clarified in Section 4.2 that when \mathbf{c} is multivariate, the \mathbf{y} -conditioned cohort is constructed with respect to all confounders

“We train $\mathbb{C}\mathbb{P}$ only on a \mathbf{y} -conditioned cohort consisting of subjects whose target \mathbf{y} values are uncorrelated with all k confounders.”

This clarification does not impact experiments where the \mathbf{y} is binary (or multinomial) as \mathbf{y} can be conditioned within one group such as in the HIV and NCANDA experiments. In the

case that \mathbf{y} is continuous (such as bone age), the \mathbf{y} -conditioned cohort needs to be assembled in such a way that their \mathbf{y} values are uncorrelated with respect to the confounder, a procedure described in Section 2.3

“the \mathbf{y} -conditioned cohort could not be defined with respect to a fixed prediction outcome. Instead, we applied the $\mathbb{C}\mathbb{P}$ component to a matched dataset, where the 3,914 boys had the same age range as the 3,518 girls (i.e., $y \in [75 \text{ months}, 175 \text{ months}]$)”

When performing this procedure with respect to multiple confounders, the sample size of the resulting \mathbf{y} -conditioned cohort is likely to reduce significantly, which reduces the quality of training of $\mathbb{C}\mathbb{P}$. A potential solution to this issue is to define a separate \mathbf{y} -conditioned cohort for each confounding variable assuming that two confounders are conditionally independent with respect to \mathbf{y} . We added this point to the limitation section:

“In case predictions are biased by multiple confounders, we would need to extend $\mathbb{C}\mathbb{P}$ to predict multiple outputs (one for each confounder) or add for each confounder a $\mathbb{C}\mathbb{P}$ component to CF-Net. In the simple scenario that the confounding variables are conditionally independent with respect to \mathbf{y} , each $\mathbb{C}\mathbb{P}$ component can be trained on a separate \mathbf{y} -conditioned cohort uniquely defined for each confounder. However, theoretical and practical ways in modeling high-order interactions between confounders require further investigation.”

Q1-2. Why do you choose the correlation scores instead of the log-likelihood, which is commonly used in prior works, e.g., [4, 54]? How does the performance changes if we change the optimization metrics?

Response: We clarified in the Discussion that log-likelihood as the adversarial loss only works for binary or categorical variables [4,54] while the correlation loss also works for continuous ones.

“Another important property of CF-Net is its ability to model continuous confounders (e.g. age) whereas most existing fair machine learning methods [9, 7, 4, 14, 61, 48] are confined to binary or discrete confounders (e.g., gender).”

Specifically, the log-likelihood for continuous variables is defined by the Mean-Squared Error (MSE), which is an ill-posed objective according to [3]. This was also supported by Supplement Table S1, Table S2, Fig. S3, and Fig. S8 that include the results based on Sadeghi et al. 2019, which used MSE as the adversarial loss. Those results indicated that MSE was less effective in removing the confounding effect in the feature space as the distance correlation and mutual information reported for Sadeghi et al. 2019 were higher than those for the correlation loss. As it is an important point of discussion but not the focus of the main article, we briefly mention the statistical properties of the correlation loss in the Discussion:

“This improvement is achieved by our novel loss function based on squared correlation (see Methods section). As discussed in our technical report [3], our adversarial loss theoretically achieves statistical mean independence between confounder and the learned features [3] and outperformed other state-of-the-art deep models in learning impartial features and unbiased model interpretation”

Q2-1. Why do you compare the performance on (a) **c**-dependent cohorts and (b) **c**-independent cohorts? The performance degradation is natural since the number of training datasets is decreased. So, I don't think that the performance drop directly indicates the CNN is substantially biased by confounder.

Response: As we now clarify in the method section, a drop in accuracy is not natural as we first trained the model on the training data set and then used the same trained model to produce two accuracy scores on the testing data: one on the whole testing data and one on a subset where the **y** values were matched with respect to **c** (we renamed it as **c**-independent subset in this revision). We clarified this point in Section 2.1

“The prediction accuracy on the testing folds was measured by balanced accuracy (BAcc) [38] (to account for different numbers of subjects in each cohort), and precision and recall rates according to the uninformative operating point of 0.5. To investigate if the prediction of the models was confounded by age, we also recorded the three accuracy scores of the approaches (without re-training) on a ‘confounder-independent’ subset (**c**-independent).”

With respect to the motivation behind comparing the accuracy on the whole cohort to the **c**-independent subset, the manuscript now clarifies in the Discussion section that we use this comparison to quantify the severity of confounding effects in a model

“by measuring the difference between the testing accuracy recorded on the whole (confounded) cohort and on the **c**-independent (unconfounded) subset. We viewed this difference as a metric for the severity of the confounding effects: the larger the difference, the more confounded the model.”

Q2-2a. Related to the Q2-1, I do not fully understand what metric the paper try to maximize/minimize, and what application scenario they are considering. In other words, it is not clear to me why the authors chose the specific metrics reported in this paper. As discussed in the fair representation learning articles, which tackle related technical problems, there are many possible metrics to compare the performance. As the different metrics measure the different aspects, I strongly recommend adding a discussion about why we need to focus on the metrics reported in this paper. Ref: “21 fairness definitions and their politics”, <https://fairmlbook.org/tutorial2.html>

Response: In line with the previous comment, we deliberate on the concept of ‘fairness’ in the Discussion. As pointed out in the reference provided by the reviewer, there is no consensus on the strict definition of fairness. One common strategy in fair learning is to inspect whether the prediction outcomes systematically differ with respect to various validation groups (i.e., concept of ‘group fairness’ or ‘demographic parity’). These validation groups can be defined based on whether the cohort is confounded or not (group fairness, e.g., whole cohort vs. **c**-independent), or can be defined with respect to subjects of different confounder values (demographic parity, e.g., young vs. old). These points were added to the Discussion

“training models on confounded data now requires evaluating the ‘fairness’ of model predictions with respect to confounders. In line with the concept of ‘group fairness’ or

‘demographic parity’ [59, 6], one can do so by examining whether the predictive power of the model varies across different ‘validation subsets’. We did so by measuring the difference between the testing accuracy recorded on the whole (confounded) cohort and on the \mathbf{c} -independent (unconfounded) subset. We viewed this difference as a metric for the severity of the confounding effects: the larger the difference, the more confounded the model. Another way of defining validation subsets is to group testing subjects according to their confounder values (see Fig. 3(b,c), 4(b-e)). In all three experiments, CF-Net achieved more balanced prediction accuracies across those subsets than ConvNet, further highlighting the fairness of the CF-Net model.”

Moreover, our evaluation extended beyond the commonly used fairness metrics that are specifically designed for binary variables and inspect confounding effects in prediction outcomes (predictive parity, equalized odds, etc). Experiments in the supplement directly quantify the confounding effects in the underlying features learned by models based on $dcor^2$, mutual information, and prediction accuracy of CP (Supplement Fig. S2, S3, S7, S8, S10). We clarified this in the discussion

“our adversarial loss theoretically achieves statistical mean independence between confounder and the learned features [3] and outperformed other state-of-the-art deep models in learning impartial features and unbiased model interpretation (see experiments in Supplement Sections A through D). These complementary tests thoroughly assessed the confounding effects in the underlying feature space and extended beyond the aforementioned fairness evaluation defined on prediction outcomes.”

Q2-2b. Related to the above comments, some statements in the introduction needs to be clarified. For example, “These different methods, however, fail to provide the means to understand the potential effect of the confounder on the outcome” why does it a problem?

Response: To clarify, we improved the overview of traditional approaches for controlling confounding effects in the introduction:

“Traditionally, studies control for the impact of confounding variables by eliminating their influences on either the output or the input variables. With respect to the output variables, one can reduce the dependency to confounders by matching confounding variables across cohorts (during data collection) [1] or through analytical approaches, such as standardization and stratification [5, 43]. Associations between confounders and input variables are frequently removed by regression analysis [2, 43], which produces residualized variables that are regarded as the confounder-free input to the prediction models.”

Q2-3. While the author only compares vanilla CNN and the proposed method, [54] can be applied for continuous variable settings. Why don’t you compare it besides?

Response: In line with the response to Q1-2, [54] only applies to binary confounders, and Xie et al. acknowledged in [54] that “... But in this paper, we focus mainly on instances where s is a discrete label with multiple choices. We plan to extend our framework to deal with continuous s

and structured s in the future.” Therefore, we performed comparison to [54] only in the bone age prediction experiment, where the confounder sex is a binary variable. The results were included in Supplement Section D and Fig. S11, S12. In a brief summary, using binary cross-entropy as in [54] produced higher prediction error than CF-Net ($p = 0.0006$, two-sample t -test).

Q2-4. How does the weighing parameter λ is selected?

Response: In the revision, we conducted additional experiments on assessing the impact of the weighing parameter λ . We first clarified in Section 4.3 that in all experiments λ was set to 1:

“We set λ to 1 (see Supplement Fig. S5)”

How the choice of hyperparameter affect the results? We performed a posthoc analysis in the HIV experiment to examine the impact of λ on classification accuracy and statistical properties of the learned features. These results were included in Supplement Section B.4 and Fig. S5.

“For each candidate $\lambda \in [0, 25]$, we performed 5-fold cross validation to record the classification accuracy on the c -independent subset and trained CF-Net on all subjects to measure $dcor^2$ on the control cohort. According to Fig. S5, small λ resulted in high $dcor^2$ values with age (large confounder effect) and low HIV classification accuracy. When using a large λ , CF-Net did not further reduce the $dcor^2$ metric but negatively impacted the accuracy of HIV classification as the model overemphasized the age prediction task in the feature-learning process. The range of $\lambda \in [0.5, 5]$ balanced classification accuracy with the constraint of conditional independence with respect to the confounder.”

We added these experiments to the supplement as we regard the hyperparameter tuning as a topic orthogonal to the study objective of examining the impact of ‘ y -conditioning’ on removing confounding effects.

Q2-4. The article describes that they used 5-fold cross-validation thorough out the paper, but never mentioned the validation procedures.

Response: In the revision, we added to Section 2.1 that

“The prediction accuracy of the models was determined via 5-fold cross-validation. On the four folds used for training, we used data augmentation to generate two cohorts of equal size ... The prediction accuracy on the testing folds was measured by balanced accuracy (BAcc) [38] (to account for different numbers of subjects in each cohort), and precision and recall rates according to the uninformative operating point of 0.5.”

- Minor comments:

- Reference on page 2 contains “?”

Response: We fixed the typo in the resubmission.

- Some parts of section 2 discuss the balance between precision and recall. However, the balance can be controlled by changing the thresholds, and therefore I’m not sure what can be said from the balance itself. Instead, PR-curve is a better metric in the case.

Response: We clarified in Section 2.1 that we set the threshold

“according to the uninformative operating point of 0.5. ”

and complemented analysis with violin plots (Fig. 3 and Fig. 4). We are of the opinion that such qualitative and quantitative description of the balance between precision and recall is scientifically rigorous. First, we avoided the need to set a data-specific threshold by creating a training set that was balanced across cohorts (by means of data augmentation). It allowed us to use a fair and non-informative threshold of 0.5 for defining precision and recall during testing. Next, we also included the violin plots in Fig. 3 and Fig 4 that reflected the balance between precision and recall more intuitively than the PR-curve. For the results of ConvNet, we highlighted the imbalance with black circles, which comprehensively outlined the skewed distribution of ConvNet prediction scores. In particular, we can see that the prediction scores of HIV subjects (or girls in the NCANDA experiment) falsely ‘leaked’ into the lower portion of the plots, supporting our discussion on the imbalanced precision and recall in the tables. For example, in Section 2.1 of the revision, we clarified

“As indicated by the black circles in Fig. 3(b), most of the young HIV subjects were falsely labelled as controls by ConvNet (only 36.3% recall rate according to Table 1) as the control cohort was significantly younger than the HIV positive subjects.”

- Why the performance on the **c**-independent cohort does not drop for the proposed method, even though the number of total examples is decreased?

Response: In line with the comment in Q2-1, the prediction accuracy does not drop as the **c**-independent subset is only used in the testing stage as clarified in Section 2.1

“To investigate if the prediction of the models was confounded by age, we also recorded the three accuracy scores of the approaches (without re-training) on a ‘confounder-independent’ subset (**c**-independent).”

One would expect a drop in accuracy if the number of training samples would decrease, which was not the case in our experiments. In other words, accuracy scores on the **c**-independent subset and whole cohort were derived by the same trained model.

- While the authors claim that “CF-Net could alleviate the confounding effect in the prediction according to the higher accuracy measured on a ‘**c**-independent’ sub-cohort compared to the ConvNet” (on the top of page 7), I do not think we can make this conclusion only from this results, as there is no confounding effect when we use the **c**-independent sub-cohort. Please clarify the statements.

Response: As mentioned in the prior response, we reworded this statement in the Discussion. We wanted to highlight that while CF-Net resulted in similar accuracy on the whole cohort and **c**-independent subset, a confounded model (ConvNet) would show reduced testing accuracy on the **c**-independent subset compared with the whole cohort.

“training models on confounded data now requires evaluating the ‘fairness’ of model predictions with respect to confounders. In line with the concept of ‘group fairness’ or ‘demographic parity’ [59, 6], one can do so by examining whether the predictive power of the model varies across different ‘validation subsets’. We did so by measuring the difference between the testing accuracy recorded on the whole (confounded) cohort and on the \mathbf{c} -independent (unconfounded) subset. We viewed this difference as a metric for the severity of the confounding effects: the larger the difference, the more confounded the model.”

- p9, line 280 needs a comma after c_ρ .

Response: We fixed the typo in the resubmission.

- While the equations do not contain expectations over the batch, I think the proposed method also uses mini-batch training. Please correct the equations.

Response: We view the mini-batch gradient descent algorithm as a numerical optimization approach that does not affect the formulation of the objective function. In line with most existing deep learning works, we define the objective function with respect to the whole cohort and use mini-batches only during optimization. We clarify this point in Section 4.3 that

“In each iteration, we first train L_p on a mini-batch sampled from all available training data. The loss of L_p was back-propagated to update θ_{fe} and θ_p . With θ_{fe} fixed, we then minimize L_{cp} to update θ_{cp} by computing the correlation of Eq. (2) over subjects of a mini-batch sampled from the \mathbf{y} -conditioned cohort. Finally, with θ_{cp} fixed, L_{cp} is maximized by updating θ_{fe} with respect to the correlation loss defined on a mini-batch from the \mathbf{y} -conditioned cohort.”

Section 4.3 should contain optimization detail for reproduction.

Response: In addition to the above optimization procedure added to Section 4.3, we also specified

“We set λ to 1 (see Supplement Fig. S5) and use a batch size of 64 subjects and Adam optimizer with a learning rate of 0.0002.”

Response to Comments from Reviewer #2:

We thank the reviewer for the positive feedback. We carefully addressed the comments. Please see the responses below

There is a lack of consistency between the three experiments in terms of exactly what results are reported. The don't include the whole-cohort result from the NCANDA experiment in Table 2, and there isn't even a results table for the bone-age experiment. For example, I think it would be important to know whether the differences in MAE reported in Figure 5d are significant. Also, the t-SNE and distance correlation results are only reported for the HIV experiment. To me, it makes sense that all three experiments are reported in a consistent way, unless there's a clear rationale for omitting some metrics in certain cases.

Response: While in the original submission we chose different representation styles to focus on the unique aspects of each experiment, we understand now that this might make the article harder to follow. In the revision, we unified the results between HIV and NCANDA experiments by adding the 'whole-cohort' results to Table 2, adding the t-SNE and saliency-map results to Figure 4, adding the comparison with the two other baseline approaches in the supplement (see Fig. S3, Fig. S8, Table S1, and Table S2). However, the setup of the bone age experiment was substantially different from the HIV and NCANDA experiments (classification vs. regression, binary vs. continuous confounder), thereby requiring the presentation of the results to differ from the previous two experiments. For instance, while the classification results can be summarized in a table of BAcc, precision and recall scores, a more intuitive way of reporting regression results are boxplots of absolute errors.

More fundamentally, I'm not sure I understood the logic of the Results. My initial assumption was that the CF-Net was designed to be beneficial (i.e., should outperform the ConvNet) when using confounded data, and that the performance would be equivocal when using the confound-independent data (in other words, when there is no confound, the CF-Net isn't necessary). However, the authors' logic seems to be the opposite of that, as in the Discussion they say "CF-Net could alleviate the confounding effect in the prediction according to the higher accuracy measured on a 'c-independent' sub-cohort compared to the ConvNet." Perhaps I'm getting the wrong end of the stick here. Is the idea that the performance in the whole cohort and confound-independent cohorts should be the same if confounds are properly accounted for? But if not controlled for (i.e., ConvNet results), the performance in whole cohort is artificially high because of the confound? I think it is essential this should be clarified (both to me and in the text), as the key results hinge on this point.

Response: The reviewer's understanding is correct. If the confounder was correctly modeled during training, the network would record similar testing accuracy on the whole and **c**-independent subset (as in the CF-Net results). If improperly modeled, the prediction accuracy would substantially drop for the **c**-independent subset (as in the ConvNet results). This finding is discussed in Section 2.1

"Compared to the whole cohort, CF-Net recorded a similar BAcc of 74.2% on the **c**-independent subset, which was significantly higher (two tailed $p = 0.035$, DeLong's

test) than the BAcc of ConvNet (BAcc: 68.4%).”

and Section 2.2

“Based on 5-fold cross-validation, the accuracy of ConvNet in predicting sex dropped from 90.3% across all samples to 87.3% on a **c**-independent subset.”

To further clarify the underlying logic, we also rephrased the Discussion:

“training models on confounded data now requires evaluating the ‘fairness’ of model predictions with respect to confounders. In line with the concept of ‘group fairness’ or ‘demographic parity’ [59, 6], one can do so by examining whether the predictive power of the model varies across different ‘validation subsets’. We did so by measuring the difference between the testing accuracy recorded on the whole (confounded) cohort and on the **c**-independent (unconfounded) subset. We viewed this difference as a metric for the severity of the confounding effects: the larger the difference, the more confounded the model.”

Results. In the confounded HIV dataset (i.e., the whole cohort), presumably the DeLong’s test was not significant? This should be reported as it suggested that CF-Net did not outperform the ConvNet in this instance. Perhaps add 4 DeLong test results to Table 1 so that it’s clear when there’s a difference between the CF-Net and the ConvNet. Would it be worthwhile using the DeLong test for all three experiments in fact?

Response: According to the reviewer’s suggestion, we computed p-values for all three experiments, which included adding DeLong’s test results to Table 1 (HIV experiment) and Table 2 (NCANDA experiment). The improvement of CF-Net compared with ConvNet on the whole cohort of the HIV experiment was at a trend-level ($p=0.068$). We added in Section 2.1 that

“Although this improvement was only on a trend-level according to DeLong’s test (two tailed $p = 0.068$), CF-Net recorded a more balanced precision (73.4%) and recall scores (75.4%) than ConvNet.”

Furthermore, CF-Net was significantly more accurate than ConvNet on the **c**-independent subset

“CF-Net recorded a similar BAcc of 74.2% on the **c**-independent subset, which was significantly higher (two tailed $p = 0.035$, DeLong’s test) than the BAcc of ConvNet (BAcc: 68.4%).”

On the NCANDA experiment, CF-Net conditioned on boys recorded significantly more accurate prediction than ConvNet during the early pubertal stage. As stated in Section 2.2:

“...was significantly more accurate in prediction at the early pubertal stage (two-tailed $p = 0.039$ DeLong’s test)...”

Note, the bone-age prediction task was a regression problem, so we report two-sample t-test results of the absolute prediction errors, i.e., Section 2.3

“the prediction accuracy of CF-Net with \mathbf{y} -conditioning was significantly higher (absolute error 11.2 ± 8.7 months) than that of the baseline ConvNet and CF-Net without \mathbf{y} -conditioning (two-tailed $p < 0.001$, one-sample $t_{3152} = 14.2$, Fig. 5(d))”.

In the Abstract, it says that the authors’ method results “in superior prediction accuracy compared to the baseline and recent invariant feature learning frameworks”. The results of the comparison with recent invariant feature learning frameworks seems to only appear in the Supplementary material and are not mentioned in the main text, unless I missed it. I recommend that anything that is mentioned in the Abstract should also appear (even if only briefly) in the main text.

Response: We agree with this comment and removed the part “and recent invariant feature learning frameworks” in the abstract. The revised main manuscript focuses on discussing the \mathbf{y} -conditioning strategy in handling the inherent correlation between features and confounders encountered in some medical applications. We also kept the comparison between our correlation loss and losses proposed by other invariant-feature-learning approaches in the supplement.

Also, on scrutiny of Table S1, I’m not convinced that the CF-Net significantly outperforms the methods by Zafar et al., or Sadeghi et al., in HIV classification, so this claim requires some further information to back it up.

Response: To clarify what we mean by the ‘superiority’ of a model with respect to our experiments, we now state in the Discussion that it

“Therefore, the superiority of a prediction model for medical imaging applications should be defined with respect to its predictive power and impartiality to confounders.”

In line with the previous comment, we also clarified in the abstract that we aimed to show that

“our method can accurately predict while reducing biases associated with confounders.”

We also added the comparison to the Zafar and Sadeghi papers to the NCANDA experiment (These two approaches do not apply to the bone-age experiment). While the prediction accuracies recorded for CF-Net in the HIV and NCANDA experiments were not significantly higher than those of the two other approaches, its features were significantly less confounded ($p < 0.001$ two-sample t -tests) than the other two approaches according to the $dcor^2$ and MI metrics reported in Table S1 and S2, Fig. S3 and S8 of the supplement.

Figure 3 f and g. The authors interpret these patterns of saliency and as being age-related and HIV-related respectively. I’m not sure that I agree with this interpretation as both age and HIV affect brain structure in a more distributed way than is apparent in the figure. Furthermore, the regions mentioned in the Results section (e.g., cerebellum) are not visible at in the figure. To make a more convincing argument that the saliency maps for the different networks do indeed

resemble ageing or HIV I think at the very least they need to include several more brain slices. To go further, it would be straightforward to conduct a VBM analysis of group age and of HIV group using the same data, then see whether the voxels where volumetric differences are associated with these factors are similar to the saliency maps.

Response: We added both sagittal and axial view of the saliency map in Fig. 3 and Fig. 4. We also added the lightbox view (multiple axial slices) of the saliency maps to the Supplement Fig. S4 and Fig. S9. Lastly, we conducted a VBM analysis to identify the HIV effect within the **c**-independence subset. The test procedure were described in the Supplement Section B3

“To put these results in perspective, we identified significant tissue loss in HIV patients by conducting voxel-based morphometry analysis [5]. Based on the MR preprocessing pipeline described in the main article, tissue classification was performed by Atropos [2] resulting in Gray-Matter (GM), White-Matter (WM), and CerebroSpinal Fluid (CSF) masks for each T1w-MRI. The GM and WM masks were non-rigidly aligned to the SRI24 atlas space by registering the T1W image to a template, corrected by Jacobian determinant of the resulting deformation, and underwent Gaussian smoothing with an FWHM of 10mm. The HIV effect was tested on each voxel in the GM masks of the **c**-independent subset by a General Linear Model implemented in Permutation Analysis of Linear Models (PALM, with 5,000 permutations) [10]. Covariates of GLM included sex, age and the diagnosis label. The resulting one-tailed voxel-wise p-values associated with the diagnosis label were corrected for spatial coherence by FSL Threshold-Free Cluster Enhancement (TFCE) [8] and for family-wise error at the 5% level across space. This test procedure was then repeated on the white-matter masks. Fig. S4c displayed voxels with significant GM (blue) and WM loss (yellow).”

and the results in Section 2.1

“Other regions with high average saliency according to CF-Net are located in the temporal lobe, inferior frontal gyrus, and subcortical regions including the amygdala and hippocampus. These regions (except for the amygdala) also exhibited significant white-matter tissue loss due to HIV according to a traditional voxel-based morphometry analysis [35] (Supplement Fig. S4)”

While findings from group statistical test (VBM) and machine learning might overlap, they generally will differ as they essentially ask different questions. VBM is a univariate analysis that answers the question of ‘whether there is a difference between groups at each voxel’, while learning models focus on multivariate patterns with large effect size to accurately infer group assignment on an individual level. Due to their distinct nature, these two approaches often result in complementary findings [Adeli et al. 2019].

[Adeli et al. 2019] Adeli, Ehsan, Natalie M. Zahr, Adolf Pfefferbaum, Edith V. Sullivan, and Kilian M. Pohl. “Novel machine learning identifies brain patterns distinguishing diagnostic membership of human immunodeficiency virus, alcoholism, and their comorbidity of individuals.” *Biological Psychiatry: Cognitive Neuroscience and Neuroimaging* 4, no. 6 (2019): 589-599.

In the Discussion they mention that they ran an SVM on FreeSurfer features on the HIV dataset and achieved 69.5%. They say that this is significantly lower than the deep models, but it's not that much lower than the ConvNet (accuracy = 71.6%) and they can only really say 'significant' if they tested that difference explicitly (e.g., DeLong's test).

Response: While the prediction accuracy scores associated with SVM were lower than the deep learning models, we agree with the reviewer that the accuracy improvement from 69% (best results from SVM methods) to 71% (ConvNet) were not statistically significant, so that we omitted that word from that sentence:

“For example, the ConvNet baseline reported a higher accuracy (BAcc: 71.6%) in the HIV experiment than applying a traditional SVM classifier to the 298 brain regional measurements extracted by FreeSurfer [18] (BAcc: 69.5%).”

Moreover, I contend that this is not really a fair comparison. FreeSurfer ROIs are by design reductive, and it's quite possible to use whole brain voxelwise data (e.g., VBM maps) as input in an SVM. My guess is that this would achieve better performance as more of the original signal has been retained. I realise that the authors' statement in the Discussion is something of an aside, but it really should be backed up by more substantive evidence or removed.

Response: According to the suggestion, we performed SVM analysis on VBM maps. We tested both linear and non-linear SVM on 4 types of voxel-wise brain maps: WM and GM maps (input to VBM analysis), Jacobian determinant maps measuring the volume change of the non-rigid deformation from the T1W space to SRI24 atlas space, raw image intensities (z-scores) of T1W images non-rigidly registered to the SRI24 atlas without downsampling, and the raw intensities of images affinely registered and downsampled (input to deep models). We found that linear SVM resulted in higher prediction accuracy than non-linear SVM. The BAcc scores for the 4 input types were 63.6%, 69.4%, 60.1%, and 63.4%, which were all lower than the BAcc reported in the manuscript (including SVM based on Freesurfer Scores). This was to be expected because SVM and other traditional machine learning approaches are more often applied to brain measurements and rarely applied to raw brain intensities (or brain maps) due to the curse of dimensionality. Therefore, we chose to not include these new results in the manuscript, but are happy to add them if reviewers or the editor are of a different opinion.

Methods – the MRI volumes for the HIV dataset were re-scaled to 64^3 . This will naturally reduce the amount of information contained in each image. How did the authors arrive at 64^3 as the right input, and what sort of interpolation was done? Presumably they did the same re-scaling to the NCANDA data?

Response: The specific resolution of $64 \times 64 \times 64$ was also chosen in other studies [Louis et al. 2019] and had been evaluated in our own prior study on 3 other neuroimaging applications [37]. The downsampling was based on spline interpolation, and the preprocessing of the NCANDA images was the same. Note, downsampling the input resolution is a common strategy in medical imaging studies to increase training batch sizes (especially for the memory-demanding 3D images) and reduce the chance of overfitting given limited training samples in this domain. These insights are now part of the Method section

“The registered images were then down-sampled to a $64 \times 64 \times 64$ volume [65] based

on spline interpolation to reduce the potential overfitting during training and to enable a large batch size”

and

“Procedures for preprocessing, downsampling, and classifying the MRI were conducted according to the HIV experiment.”

[Louis et al. 2019] Louis M., Couronné R., Koval I., Charlier B., Durrleman S. (2019) Riemannian Geometry Learning for Disease Progression Modelling. In: Information Processing in Medical Imaging. IPMI 2019.

Figure 5c – I’m not clear what exactly has been plotted here. Can the authors please clarify? Generally, I suggest not to use bar plots for anything other than count data and I think the y-axis is continuous here.

Response: We changed the bar plot to a curve plot. We also clarified in Section 2.3

“The model tended to overestimate the age of girls compared to boys (Fig. 5(b)), and this discrepancy was more pronounced in the age range of 110 to 200 months.”

and in the caption of Fig. 5

“ConvNet tended to predict higher age for girls than boys indicating a confounding effect of sex; (c) This prediction gap between boys and girls was more pronounced in the age range of 110 to 200 months but was significantly reduced by CF-Net”

Did the authors consider visualising the confounding effects from the adversarial component of the network (i.e., CP)?

Response: As stated in the new limitation section, visualizing confounding effects within the proposed CF-Net is a topic of research. As shown in Supplement Fig. S2, S7 and S10, the learned features in CF-Net are no longer predictive of confounder values, so the CP component could not be used as a valid predictor (i.e., CP was already ‘fooled’ by the adversarial training). Therefore, the saliency visualization based on back-propagation [29] is not sensible. We acknowledged this in the limitation section

“While we were able to visualize the HIV and sex effect by computing saliency maps [29] inferred from the predictor \mathbf{P} , the same technique is not directly applicable to visualize confounding effects from CP due to the adversarial training. An alternative could be deriving saliency maps from CP retrained on the features learned by the baseline ConvNet (e.g., Supplement S2), i.e., a model that substantially captures the confounding effect.”

To what extent does the data augmentation scheme (especially the left-right flipping) influence the saliency maps?

Response: As described in the Supplement Section B.3, we only used data augmentation and flipping during training, and the saliency maps were derived from raw images based on the learned model. This procedure is now detailed in Section 4.1

“Lastly, a saliency map was computed [29] for the right hemisphere of each test image quantifying the importance of each voxel to the final prediction.”

and in Supplement B.3

“After each training run, we computed the saliency map for the right hemisphere of each test image (without augmentation and flipping) based on the model learned on the training folds.”

We chose to perform left-right flipping during training based on the understanding that the HIV effects and sex differences in the adolescent brain were bilateral [41,1]. This assumption was supported by the similar accuracy scores when training models on images without flipping (ConvNet: 71.6%, CF-net: 74.1%) compared to those of Table 1. We did not include these numbers in the manuscript as they were only marginally relevant. However, not flipping will result in the model randomly focusing on one hemisphere more than the other, as learning algorithms tend to discard redundant information when correlated features (such as from left and right hemispheres) are present. Interpreting the outcome of the model will thus be more complicated.

The precise construction of the Feature Extraction (FE) network is likely to have a big impact on exactly what features get used in the model. For bone age they used VGG, but for the MRI experiments the FE network was trained from scratch. How did they decide on the specific configuration on the network?

Response: We clarified in Section 4.3 that we used VGG which is one of the most widely used architectures for 2D images.

“For the 2D X-ray experiment, the FE and P components complied with the feature extractor and predictor defined in [34]”

For the MRI experiments, we clarified in Section 4.3 that

“With respect to the network architecture used in the experiments, we followed the design of FE in [37,65] ...”

It is conceivable that incorporating more advanced network components could potentially further improve the prediction accuracy. However, we aimed to conduct experiments based on simple network architectures in this study to ensure generalizability of the results. $2 \times 2 \times 2$ convolution/max-pooling, batch-normalization and ReLU activation are some of the most fundamental and commonly used blocks in ConvNets. Similar network architectures achieved reasonable results in our prior studies [37] on three different neuroimaging applications. The exploration of network architecture is now briefly discussed in the limitation section as a future direction.

“Finally, we abstained from determining the optimal implementation of the proposed confounder-free modelling strategy by performing extensive exploration of network architectures. Instead, we relied on some of the most fundamental network components used in deep learning. This rather basic implementation still recorded reasonable prediction accuracies so the findings discussed here are likely to generalize to more advanced network architectures.”

On a more philosophical note, I think some discussion of the following would be informative for readers. If this network can remove the influence of confounders, does this mean that there is less need to match samples for confounders (e.g., age) when recruiting? Or are we not at that stage yet?

Response: Thank you for this relevant comment. To address this, we added the following text in the discussion section:

“As discussed, the proposed CF-Net permits training deep models on cohorts not strictly matched with respect to confounders and avoids discarding unmatched samples. However, this does not mean that there is no need to keep the confounders in mind when recruiting participants for medical imaging studies. For learning models and statistical group tests, performing analysis on confounder-matched cohorts with sufficient samples (**c**-independent subsets) remains a fundamental strategy to disentangle biomarkers of interest from effects of confounders. For instance, when training a classifier to distinguish two groups that are completely biased by an age effect (e.g., one has participants with strictly larger age than the other), there is no guarantee that any method, including ours, can remove the extraneous effects of age in a purely data-driven fashion.”

Response to Comments from Reviewer #3:

We thank the reviewer for the positive feedback on the idea of our paper. We carefully addressed the comments. Please see the responses below

1. The authors proposed Confounder-Free Neural Network with a GAN-like formulation of a min-max game between the classification prediction network and the confounder prediction network. The formulation seems straightforward. It will be helpful if the authors can draw connections and highlight the differences between the proposed method and the existing methods in the literature.

Response: In response to the comment, we further deliberate on the connection and differences between CF-Net and other invariant-feature-learning approaches in the introduction and discussion.

Introduction:

“Possible alternatives could be unbiased [14, 31, 56, 28, 50] and invariant feature learning approaches [61, 19, 7], which rely on end-to-end training to study the invariance (independence) between the learned features \mathbf{F} and a bias factor (① in Fig. 2(b)). Despite the similarity in the problem setup, ignored by these methods yet of great importance to medical imaging studies is selecting features \mathbf{F} predictive of the outcome \mathbf{y} (i.e., ③ in Fig. 2(b)), while accounting for the intrinsic relationship between \mathbf{y} and the confounder \mathbf{c} (i.e., ② in Fig. 2(b)).”

“Instead of enforcing marginal independence between \mathbf{F} and \mathbf{c} as we propose in [3], a more principled way of correcting confounding effects is to only remove the direct association between \mathbf{F} and \mathbf{c} (① in Fig. 2(b)) while preserving their indirect association with respect to \mathbf{y} (② & ③ in Fig. 2(b)). We therefore specifically train $\mathbb{C}\mathbb{P}$ on a ‘ \mathbf{y} -conditioned’ cohort, i.e., samples of the training data whose \mathbf{y} values are confined to a specific range (referenced as ρ in Fig. 2(a)). In doing so, the features learned by CF-Net are predictive of \mathbf{y} while being conditionally independent of \mathbf{c} ($\mathbf{F} \perp\!\!\!\perp \mathbf{c} | \mathbf{y}$).”

Discussion:

“While recent advances in adversarial learning have shed light on this problem, existing deep models were only designed to tackle specific confounding effects such as scanner difference or dataset harmonization [27, 32, 8].”

“Another important property of CF-Net is its ability to model continuous confounders (e.g., age) whereas most existing fair machine learning methods [9, 7, 4, 14, 61, 48] are confined to binary or discrete confounders (e.g., gender).”

“general fair/invariant feature learning frameworks could potentially be harmful in this situation as it is impossible to derive features that are simultaneously discriminative with respect to \mathbf{y} and independent with respect to \mathbf{c} To address this

issue, we proposed here to learn the direct link between \mathbf{F} and \mathbf{c} by modeling their conditional independence in a \$\mathbf{y}\$ -conditioned cohort”

2. Is the proposed method CF-Net the same as the BR-Net in the authors’ prior publication titled “Representation learning with statistical independence to mitigate bias”? If so, this work’s contribution might be at the lighter end as the novel part of this manuscript is applying the previously proposed methods on three medical image applications. If not, it will be helpful to make this clear in the paper and highlight the distinctions.

Response: The short answer is that the proposed CF-Net differs from BR-Net described in [3]. [3] is an unpublished (in a non peer-reviewed venue) technical report written by us, which presents a method for mitigating the effects of biases in deep learning models. In comparison with [3] and other prior work for mitigating biases, we argue in the Introduction:

... unbiased [14,31,56,28,50] and invariant feature learning approaches [61,19,7,3] ... rely on end-to-end training to study the invariance (independence) between the learned features \mathbf{F} and a bias factor (① in Fig. 2(b)). Despite the similarity in the problem setup, ignored by these methods yet of great importance to medical imaging studies is selecting features \mathbf{F} predictive of the outcome \mathbf{y} (*i.e.*, ③ in Fig. 2(b)), while accounting for the intrinsic relationship between \mathbf{y} and the confounder \mathbf{c} (*i.e.*, ② in Fig. 2(b)). An example of such an intrinsic relationship with respect to the age-confounded MRI dataset is to distinguish the healthy aging of the brain in controls from aging accelerated by a disease, such as HIV infection [13,12,39].

As a result, all prior models for mitigating bias (including [3] and others referenced above) are generally incapable of removing effects of confounders, which our experiments verified.

Here, we further advance the invariant-feature-learning approach of [3] by proposing to enforce conditional independence based on learning confined to \mathbf{y} -conditioned cohorts. This innovation is critical for applying the approach to medical images. We have clarified these novel contributions in the revised manuscript, which is the focus of the remainder of this response.

First, instead of repeating the details presented in [3] we highlighted the advantage of that novel loss function in the Introduction

“Beyond that, the supplement summarizes additional experiments on the three data sets (Section B through D) and on a synthetic data set (Section A). These results converge with the theoretical advantages of our novel adversarial loss function (over state-of-the-art invariant feature learning schemes). As we systematically studied in the technical report [3], these advantages include the ability to handle continuous confounding variables and guaranteeing mean independence between \mathbf{F} and \mathbf{c} .”

and in the Discussion

“As discussed in our technical report [3], our adversarial loss theoretically achieves statistical mean independence between confounder and the learned features [3] and

outperformed other state-of-the-art deep models in learning impartial features and unbiased model interpretation (see experiments in Supplement Sections A through D).

With respect to motivating the need for introducing conditional independence, we point out in the Introduction that confounding variables in medical images play a different role than the protected bias variables of invariant-feature-learning discussed in [3]:

“Instead of enforcing marginal independence between \mathbf{F} and \mathbf{c} as we propose in [3], a more principled way of correcting confounding effects is to only remove the direct association between \mathbf{F} and \mathbf{c} (① in Fig. 2(b)) while preserving their indirect association with respect to \mathbf{y} (② & ③ in Fig. 2(b)). We therefore specifically train $\mathbb{C}\mathbb{P}$ on a ‘ \mathbf{y} -conditioned’ cohort, i.e., samples of the training data whose \mathbf{y} values are confined to a specific range (referenced as ρ in Fig. 2(a)). In doing so, the features learned by CF-Net are predictive of \mathbf{y} while being conditionally independent of \mathbf{c} ($\mathbf{F} \perp\!\!\!\perp \mathbf{c} | \mathbf{y}$).”

Finally, modelling confounders is still under-explored in deep learning so we state in the Introduction:

“To the best of our knowledge, this is the first attempt to design an end-to-end, confounder-free prediction model for medical images, in which the goal is not only to learn features invariant to a bias variable but also to properly model interactions among all three variables in a confounded situation.”

and in the Discussion

“general fair/invariant feature learning frameworks could potentially be harmful in this situation as it is impossible to derive features that are simultaneously discriminative with respect to \mathbf{y} and independent with respect to \mathbf{c} To address this issue, we proposed here to learn the direct link between \mathbf{F} and \mathbf{c} by modeling their conditional independence in a \mathbf{y} -conditioned cohort”

3. Since this paper’s main contribution is the proposal of the CF-Net method, further discussion on the design choices is needed, especially the choices of L_{cp} and the training of CP only on a \mathbf{y} -conditioned cohort. These choices should be justified and compared with alternative methods.

Response: We now further justify the design choices and discuss alternative methods in the response to the proceeding comments by the reviewer. Specifically, the design choice of L_{cp} is clarified in the following places:

As stated in the introduction, we offered the first solution in training end-to-end deep learning models to handle inherent correlation between the confounder \mathbf{c} and prediction outcome \mathbf{y} . We motivated the need for ‘ \mathbf{y} -conditioning’ by emphasizing the challenge that

“Despite the similarity in the problem setup, ignored by these methods yet of great importance to medical imaging studies is selecting features \mathbf{F} predictive of the outcome \mathbf{y} (i.e., ③ in Fig. 2(b)), while accounting for the intrinsic relationship between \mathbf{y} and the confounder \mathbf{c} (i.e., ② in Fig. 2(b)).”

To resolve this issue, we now clarify the proposal in the Introduction

“Instead of enforcing marginal independence between \mathbf{F} and \mathbf{c} as we propose in [3], a more principled way of correcting confounding effects is to only remove the direct association between \mathbf{F} and \mathbf{c} (① in Fig. 2(b)) while preserving their indirect association with respect to \mathbf{y} (② & ③ in Fig. 2(b)). We therefore specifically train $\mathbb{C}\mathbb{P}$ on a ‘ \mathbf{y} -conditioned’ cohort, i.e., samples of the training data whose \mathbf{y} values are confined to a specific range (referenced as ρ in Fig. 2(a)). In doing so, the features learned by CF-Net are predictive of \mathbf{y} while being conditionally independent of \mathbf{c} ($\mathbf{F} \perp\!\!\!\perp \mathbf{c} | \mathbf{y}$).”

In all three experiments, we explore different strategies of defining the \mathbf{y} -conditioned cohort. In the HIV experiment, we clarified in Section 2.1 that

“... confined the predictions of age by $\mathbb{C}\mathbb{P}$ to controls (i.e., the \mathbf{y} -conditioned cohort was defined by $\mathbf{y} = 0$).”

and explained in the Discussion section that

“In the HIV experiment, the relation between \mathbf{F} and \mathbf{c} was supposed to capture normal aging, which could only be studied on the control group (fixing $\mathbf{y} = 0$) as HIV accelerates brain aging [13, 12, 39].”

In the NCANDA experiment, we defined the \mathbf{y} -conditioned cohort separately with respect to boys and girls.

“For CF-Net, the accuracy depended on the set of subjects used for training the component $\mathbb{C}\mathbb{P}$, which, unlike in the HIV experiment, was not uniquely defined as the modelling of the PDS effect could be conditioned on $\mathbf{y} = 0$ (boys) or $\mathbf{y} = 1$ (girls).”

We found the model performed the best when conditioned on boys, and the conditioning with respect to girls negatively affected the performance. We explained this finding in the Discussion that

“In the NCANDA experiment, boys ($\mathbf{y} = 0$) or girls ($\mathbf{y} = 1$) would have been theoretically suitable to train $\mathbb{C}\mathbb{P}$ being impartial to PDS. Of the two cohorts, training conditioned on boys resulted in more impartial predictions as this cohort covered the full range of PDS values, while lower PDS scores were not well represented in the girl-conditioned cohort as adolescent girls are generally more mature than boys of the same age.”

In the bone age experiment, we also explored \mathbf{y} -conditioning with respect to a continuous \mathbf{y} variable in Section 2.3, the results of which outperformed CF-Net without conditioning (Fig. 5)

“we applied the $\mathbb{C}\mathbb{P}$ component to a matched dataset, where the 3,914 boys had the same age range as the 3,518 girls (i.e., $y \in [75 \text{ months}, 175 \text{ months}]$)”

For example, why only train on a single \mathbf{y} -conditioned cohort instead of having some training scheme that utilizes all possible \mathbf{y} -conditioned cohorts.

Response: As pointed out in the prior response, using all possible \mathbf{y} -conditioned cohorts can lead to suboptimal results such as in the case of the NCANDA experiment when modelling the PDS effect specific to girls. This was stated in Section 2.2

“Note, conditioning the modeling between \mathbf{F} and \mathbf{c} on girls not only reduced the overall balanced accuracy in each sub-group but also enlarged the discrepancy in precision and recall rates (Fig. 4(d)).”

And why using squared correlation as the loss function instead of other alternatives.

Response: We clarified that most existing adversarial losses only work for binary or categorical variables while the correlation loss also work for continuous ones. This is now clarified in the Discussion

“Another important property of CF-Net is its ability to model continuous confounders (e.g. age) whereas most existing fair machine learning methods [9, 7, 4, 14, 61, 48] are confined to binary or discrete confounders (e.g., gender).”

We also added new experiments to show that our squared correlation loss outperforms two other loss functions proposed in ‘Zafar et al. 2017’ and ‘Sadeghi et al’ that are applicable to continuous variables. These results were included in the Supplement for the HIV study in the initial submission (Table S1, Figure S3) and replicated for the NCANDA study in this revision (Table S2, Figure S8). In both experiments, the correlation loss produced features more impartial to confounders. We explained this finding in the Discussion

“This improvement is achieved by our novel loss function based on squared correlation (see Methods section). As discussed in our technical report [3], our adversarial loss theoretically achieves statistical mean independence between confounder and the learned features [3] and outperformed other state-of-the-art deep models in learning impartial features and unbiased model interpretation”

Moreover, we also added experiments in the bone-age study to show that the squared correlation loss has the potential to outperform state-of-the-art invariant-feature-learning methods even if the confounder is binary. Specifically, the comparison to the loss function proposed in “Xie 2017” was given in Supplement Fig. S10.

4. Please describe how the operating points were selected for results in Tables 1 and 2.

Response: We now clarified in Section 2.1 that

“The prediction accuracy on the testing folds was measured by balanced accuracy (BAcc) [38] (to account for different numbers of subjects in each cohort), and precision and recall rates according to the uninformative operating point of 0.5.”

We chose this fair and non-informative threshold as the sample size in the training set was balanced across cohorts (by means of data augmentation). We complemented the metrics derived from this threshold by the violin plots (Fig. 3 and 4), which intuitively outlined the whole distribution of prediction scores for each cohort.

5. Please describe the approach to select the confounder-independent cohort.

Response: As requested, we extended the description of the matching algorithm in Section 4.1

“Construction of the \mathbf{c} -independent subset was based on the matching algorithm [1] that extracted the maximum number of subjects from each group in such a way that they were equal in size and identically distributed with respect to the confounder values. For each HIV subject, we selected a control subject with minimal age difference and repeated this procedure until all HIV subjects were matched or the two-tailed p -value of the two-sample t -test between the two age distributions dropped to 0.5.”

6. Please provide confidence intervals for the results in Tables 1 and 2.

Response: Instead of reporting on the confidence interval directly, we added the p -value of DeLong’s test to Table 1 and Table 2 as we found them more informative. The most stringent way of defining confidence intervals is based on bootstrapping [16], which is computationally impractical with respect to our deep learning model as it requires at least hundreds of cross-validation runs to properly define the empirical distribution of the accuracy score (200 runs of 5-fold cross-validation correspond to 1,000 training runs). Alternative approaches are based on parametric models, such as proposed in [Witten et al. 2005], which are inferred from the sample size. As we explain in the next paragraph, these confidence intervals seem not very informative. We are happy to add them if the reviewer or editor is of a different opinion.

Following the procedure in [Witten et al. 2005], each prediction is a binary decision that can be regarded as a Bernoulli trial. The number of successful trials (correct classification) in a Bernoulli process follows a binomial distribution, which can be approximated by Gaussian for large N (for $N > 30$). In this situation, the 95% interval can be directly derived from the BAcc score and sample size: $c = 1.96\sqrt{\text{BAcc}(1 - \text{BAcc})/N}$. For instance, in the HIV experiment, the confidence interval on the \mathbf{c} -independent subset is $68.4\% \pm 5.8\%$ for ConvNet and $74.2\% \pm 5.4\%$ for CF-Net. In the NCANDA experiment, the confidence interval on the \mathbf{c} -independent cohort is $87.8\% \pm 3.3\%$ for ConvNet, $83.3\% \pm 3.7\%$ for CF-Net conditioned on all, $84.3\% \pm 3.6\%$ for CF-Net conditioned on girls, and $88.5\% \pm 3.1\%$ for CF-Net conditioned on boys.

[Witten et al. 2005] Data Mining: Practical Machine Learning Tools and Techniques, Second Edition (Morgan Kaufmann Series in Data Management Systems), Ian H. Witten, Eibe Frank (June 22, 2005)

7. L89: “Only 36.3% of the young HIV subjects were correctly labeled.” With this context, how should we understand the evaluation metrics in Table 1 given these incorrect labels?

Response: This result indicates that the prediction of ConvNet was confounded and largely based on age. Since the CTRL cohort was younger, ConvNet tended to label young HIVs also as CTRLs. Further supporting this observation was the low recall rate in Table 1 (c-independent young) and the skewed distribution of prediction scores produced by ConvNet (circled in Fig. 3(b)). We clarified in Section 2.1 that

“As indicated by the black circles in Fig. 3(b), most of the young HIV subjects were falsely labelled as controls by ConvNet (only 36.3% recall rate according to Table 1) as the control cohort was significantly younger than the HIV positive subjects.”

8. Section 2.2: Why are there asymmetric results such that conditioning on boys has different results from conditioning on girls?

Response: We now explain this finding in further detail in the Discussion. In the NCANDA experiment, the PDS effect could only be effectively modeled in boys (fixing $\mathbf{y} = 0$) as they offered a wide distribution of PDS enabling a robust regression learning by $\mathbb{C}\mathbb{P}$. Conversely, most girls attained maturity during the age span of NCANDA, so the narrow distribution of the plateaued PDS prevented further stratification by $\mathbb{C}\mathbb{P}$.

“In the NCANDA experiment, boys ($\mathbf{y} = 0$) or girls ($\mathbf{y} = 1$) would have been theoretically suitable to train $\mathbb{C}\mathbb{P}$ being impartial to PDS. Of the two cohorts, training conditioned on boys resulted in more impartial predictions as this cohort covered the full range of PDS values, while lower PDS scores were not well represented in the girl-conditioned cohort as adolescent girls are generally more mature than boys of the same age. ”

9. Discussion: it will be great if the authors can comment on how this method can be extended to scenarios without knowing the confounding variables of interest.

Response: We thank the reviewer for this valuable suggestion. Handling unknown confounders is an open research topic for both traditional learning models and statistical analysis approaches, and is scarcely explored in deep learning. In the revision, we added the following discussion to the Section of Limitations:

“A limitation of our experiments was the focus on single confounders that were known a priori. To model unknown confounders, we aim to explore coupling CF-Net with causal discovery algorithms (such as [58, 21, 52]).”

10. I don't quite understand the training approach of only training CP on a cohort of the same \mathbf{y} . Given a binary dataset with balanced $\mathbf{y}=0$ and $\mathbf{y}=1$, this is essentially only using half of the dataset to train CP. Why can't we train with both?

Response: In line with the response to Comment 3, the reasons for using only one group were different between the HIV and NCANDA experiments. We now clarify this point in the discussion:

“However, the specific group chosen to model the conditional dependency is application-specific.”

With respect to the HIV experiment, we explained in the Discussion section that

“In the HIV experiment, the relation between \mathbf{F} and \mathbf{c} was supposed to capture normal aging, which could only be studied on the control group (fixing $\mathbf{y} = 0$) as HIV accelerates brain aging [13, 12, 39]”

With respect to the NCANDA experiment, we explained in the Discussion section that

“In the NCANDA experiment, boys ($\mathbf{y} = 0$) or girls ($\mathbf{y} = 1$) would have been theoretically suitable to train $\mathbb{C}\mathbb{P}$ being impartial to PDS. Of the two cohorts, training conditioned on boys resulted in more impartial predictions as this cohort covered the full range of PDS values, while lower PDS scores were not well represented in the girl-conditioned cohort as adolescent girls are generally more mature than boys of the same age. ”

In Section 2.2 and Table 2, the lower prediction accuracy of ‘CF-Net conditioned on girls’ than ConvNet also justified not to use both \mathbf{y} -conditioned cohorts for the NCANDA experiment.

“Note, conditioning the modeling between \mathbf{F} and \mathbf{c} on girls not only reduced the overall balanced accuracy in each sub-group but also enlarged the discrepancy in precision and recall rates (Fig. 4(d)).”

11. For this GAN-like min-max training, it will be informative for providing the loss functions with breakdowns of L_p and L_{cp} over the train and tune set in the Supplementary.

Response: As requested by the reviewer, we added the loss curves for the HIV experiment in Supplement Fig. S6, and briefly discussed the curves in Supplement Section B.5

“Both loss curves approximately converged after training with 1,000 mini-batches indicating the model simultaneously achieved accurate HIV classification (low prediction loss) and confounding effect removal (low correlation loss). The slight oscillation of L_{cp} after 1,000 iterations was likely to be the result of the competing game underlying the min-max objective (Eq. 3, main article).”

I’m also curious of how CP performs in predicting confounding variables after the model converges, this will highlight to what extent can the confounding information be removed in F.

Response: As requested, in all three experiments, we trained $\mathbb{C}\mathbb{P}$ to predict the confounder value from the learned features upon the model converges. The testing procedure was summarized in Supplement Section B.1 and the results were given in Fig. S2, S7, and S10. These results indicate that in all three experiments the confounding effects were significantly reduced by CF-Net in the feature space. Further, in the HIV experiment the prediction accuracy of $\mathbb{C}\mathbb{P}$ was not significantly better than that of the null classifiers (that produce random or uniform predictions), indicating that the confounding effects were fully removed from the feature space (Supplement Fig. S2). This result is summarized in Section 2.1

“To assess that the unbiased prediction of CF-Net was the result of extracting features impartial to normal aging, we performed a post-hoc analysis, in which we trained CP to predict age from the learned features. Upon convergence of the training loss in each run of the 5-fold cross-validation, the post-hoc analysis re-trained CP from scratch on the features extracted from the controls in the training folds and recorded the predicted age of the controls in the testing fold. According to Supplement Fig. S2, the features learned by CF-Net no longer contained aging information as the prediction of age was nearly random (Pearson’s $r = 0.12$, two tailed $p = 0.17$). However, training CP on the features learned by 3D ConvNet resulted in age prediction of significant accuracy (Pearson’s $r = 0.95$, two tailed $p < .0001$).”

Minor comments:

1. L95: upper left region

Response: We fixed this typo in the revision.

2. Section 2.1: If I understand correctly, the proposed CF-Net is compared against the same architecture but without the CP part. It might be easier to make this clear and rename the ConvNet as ConvNet. 3D isn’t the focus, and if you need to add 3D to the name, you might also want to update CF-Net to 3D CF-Net.

Response: As requested, we now refer to the baseline as ConvNet instead of 3D ConvNet.

L160: Can you elaborate a bit more on why a more localized saliency map is preferred in predicting age from X-ray?

Response: To address this comment, we modified and clarified the sentence in Section 2.3 as:

“The saliency maps of CF-Net were more localized on anatomical structures than those of ConvNet indicating that the widespread pattern leveraged by ConvNet might be redundant and relate to confounder-related cues.”

Reviewers' Comments:

Reviewer #1:

Remarks to the Author:

Thank you for providing detailed response. I took a look at the revised paper, author response, and comments from other reviewers. As a whole, the revision makes the manuscripts clearer. However, I still have several concerns and questions.

(1) Regarding the y -conditioning procedures for regression task.

According to the revision, the authors define y -conditioning cohort for continuous prediction tasks as data confined to an interval (e.g., age range). However, the confounder effect could be remaining after this process because it does not necessarily ensure the decorrelation between c and y , without carefully selecting the interval.

(1-1) How the selection quality effect the performance?

(1-2) As the strict range selection possibly significantly decrease the sample size, there are tradeoff between sample size and the quality of cohort. For example, when picking up only one values, the cohort is strictly y -conditioned, but sample size might be small. How can we select the proper interval in practice?

(2) > our adversarial loss theoretically achieves statistical mean independence between confounder and the learned features [3]

This claim is very weird for me, because [3] does not give any theoretical guarantee when y -conditioning techniques are incorporated. Please clarify the sentence.

(3) > we used data augmentation to generate two cohorts of equal size and conned the predictions of age by CP to controls (i.e., they-conditioned cohort was defined by $y = 0$)."

(3-1) Why do we need to generate two cohorts of equal size?

(3-2) What data augmentation did you use (for reproducibility) ?

<Recommendations>

Below are several recommendations for authors to make a paper strength or clear, while it it not requirements for acceptance.

(3) Methodological perspectives.

(3-1) Baseline methods

The main manuscripts compare the ConvNet and CF-Net, however, the outperforming the convnets is somewhat trivial as suggested by many other invariant feature learning papers. In other words, from the current main manuscripts, the contribution of y -conditioning itself, which is the main methodological contributions, is not clear. On the other hands, the supplemental materials contain several comparisons with more reasonable baselines, which might be more interesting for potential readers. I recommend authors to move some comparison between more reasonable methods into main manuscripts.

(3-2) Regarding [54].

As, the authors clarified, [54] did not apply the method the case where the confounders are continuous. However, the formulation of [54] itself is general enough for continuous case. This is also clearly mentioned in [54] as below (and as the authors may acknowledge).

"Note that under our framework, in theory, s can be any type of data as long as it represents an attribute of x . For example, s can be a real value scalar/vector, which may take many possible values, or a complex sub-structure such as the parse tree of a natural language sentence. But in this paper, we focus mainly on instances where s is a discrete label with multiple choices."

As the authors mentioned in the response, the log-likelihood of continuous variables is defined by the mean squared error, and the [54] can be easily applied for continuous case. Thereby, the extensive comparison with [54] is possible, and the results might strengthen the paper.

(4) The impact for medical practitioners.

In my opinion, the potential impact for medical practitioners are not clear from the current manuscripts. For example, when the practitioner should use the proposed method? How can we incorporate the domain knowledge (if possible)? To make the broader impact, such a discussion is preferable.

Reviewer #2:

Remarks to the Author:

The authors have done a good job responding to the comments from myself and the other reviewers and the manuscript is markedly improved as a result. I have no further comments or suggestions.

Reviewer #3:

Remarks to the Author:

Thanks for the detailed response to my previous comments. I have one remaining comment. For the added DeLong tests in Tables 1 and 2, please provide the p-values in the Supplementary materials. The authors used 0.05 as the significance threshold for p-values. However, given the number of tests the authors performed, the authors should correct for multiple hypothesis testing. In Tables 1 and 2, 3 out of 8 tests with p-values slightly less than 0.05 are precisely the scenario where a correction is needed.

Submission ID: NCOMMS-20-23428A

Training Confounder-Free Deep Learning Models for Medical Applications

We appreciate the valuable comments and feedback from the reviewers. Point-to-point responses to the reviewers' comments and the modifications in the manuscript are listed in the following. We also provide a version of the revised manuscript, in which all the modified text are typeset in blue.

Response to Comments from Reviewer #1:

We thank the reviewer for the positive feedback on our revision. We carefully addressed the new comments. Please see the responses below

Thank you for providing detailed response. I took a look at the revised paper, author response, and comments from other reviewers. As a whole, the revision makes the manuscripts clearer. However, I still have several concerns and questions.

(1) Regarding the y -conditioning procedures for regression task. According to the revision, the authors define y -conditioning cohort for continuous prediction tasks as data confined to an interval (e.g., age range). However, the confounder effect could be remaining after this process because it does not necessarily ensure the decorrelation between c and y , without carefully selecting the interval.

Response: We further clarify in Section 2.3 and Section 4.1 of the revised manuscript that the decorrelation between age and sex was not due the specific interval selection but guaranteed by a bootstrapping procedure performed on the interval.

In Section 2.3, we clarified

“Instead, we applied the CP component to a bootstrapped training set of 10,000 boys and 10,000 girls whose age was confined to the interval from 75 months to 175 months and had strictly matched distributions between the two genders (see Methods section).”

In Section 4.3, we expanded the description of the bootstrapping procedure

“3,914 boys and 3,518 girls, or 80% of the training subjects (Fig. 5a), had bone ages between 75 months and 175 months (the Full Width at Half Maximum of the age distribution, Supplement Fig. S10). Confined to this age range, we used bootstrapping [18] to generate 1,000 boys and 1,000 girls within each 10-month interval. This procedure resulted in a y -conditioned cohort of 10,000 boys and 10,000 girls strictly matched with respect to bone age ($p = 0.19$, two-tailed two-sample t -test).”

(1-1) How the selection quality effect the performance?

Figure R1: Extension of Fig. 5 of the main text: (a) Prediction accuracy of CF-Net based on interval-based or matching-based \mathbf{y} -conditional cohorts; (b) Discrepancy in predicted age between boys and girls (sex effect) for each approach.

Response: To address the reviewer’s concern, we compared the results of CF-Net based on two constructions of the \mathbf{y} -conditional cohort. The first one, denoted here as “interval-based CF-Net”, was the \mathbf{y} -conditioned cohort defined in the main text, which was derived from a pre-determined age interval. The second \mathbf{y} -conditioned cohort, denoted here as “matching-based CF-Net”, was derived by an entirely data-driven matching procedure applied to all ages of the data. We found that the two CF-Nets performed similarly with respect to prediction accuracy and confounding effect removal. We now describe this finding in more detail:

For matching-based CF-Net, the data-driven matching^[1] first constructed a bipartite graph such that the first set of nodes represented 5,415 boys and the second set of nodes represented 4,313 girls. An edge was created between any girl and boy with an age gap smaller than 6 months. A Ford-Fulkerson algorithm^[2] was then applied to select the maximum number of matching pairs. This process resulted in an age-matched dataset of 3,730 boys and 3,730 girls (age difference: $p = 0.21$ two-sample t -test).

Trained with respect to this \mathbf{y} -conditioned cohort, the prediction accuracy of CF-Net (absolute error 11.2 ± 9.3 months) was similar to the interval-based CF-Net (absolute error 11.2 ± 8.7 months; see Section 2.3 of the main text) according to a two-sample t -test ($p = 0.15$). Compared to the three baselines, both CF-Nets were significantly more accurate ($p < 0.001$, Figure R1a) and reduced the discrepancy in predicted age between boys and girls (Figure R1b).

Given the similar performance between the two constructions of the \mathbf{y} -conditional cohort, we briefly discussed the new finding in the discussion section:

“When predicting a continuous variable, we proposed to define the \mathbf{y} -conditioned co-

Figure R2

hort by selecting samples whose \mathbf{y} was confined to a fixed interval and decorrelating \mathbf{y} and \mathbf{c} via bootstrapping. In the bone age experiment, the interval was selected as the Full Width at Half Maximum (FWHM) [60] of the overall age distribution, which approximately encompassed 80% of the training subjects and focused only on the age range with sufficient samples (Supplement Fig S10). This well-represented age interval facilitated the decorrelation with respect to gender and resulted in a large \mathbf{y} -conditioned cohort for training CP. Another strategy for defining the interval (not explored in this article) is to model the interval as a hyperparameter, whose optimal setting is determined via parameter exploration during nested cross-validation. Alternatively, one can bypass the need of selecting interval by using data-driven matching procedures (e.g., a bipartite graph matching [51] or greedy algorithm [1]), which in our experiments produced similar accuracy scores as the one based on the FWHM criteria and bootstrapping. ”

[1] Rosenbaum PR. Optimal Matching for Observational Studies. *Journal of the American Statistical Association*. 1989;84(408):1024–1032.

[2] Ford LR, Fulkerson DR. Maximal flow through a network. *Canadian Journal of Mathematics*. 1956;8:399-404.

(1-2) As the strict range selection possibly significantly decrease the sample size, there are tradeoff between sample size and the quality of cohort. For example, when picking up only one values, the cohort is strictly \mathbf{y} -conditioned, but sample size might be small. How can we select the proper interval in practice?

Response: As we discuss below and in the revised discussion, there are several strategies for selecting the proper interval.

With respect to the strategy chosen in this manuscript, the \mathbf{y} -conditioned cohort was determined by first selecting an age interval according to the Full Width at Half Maximum (FWHM) [60] of the age distribution (Figure R2) to balance between the size of the \mathbf{y} -conditioned cohort

(width of age interval) and quality of decorrelation (bootstrapping). Choosing a smaller interval (e.g., the ‘one sample’ example mentioned by the reviewer) increases the risk of overfitting. On the other hand, if one aims to maximize the sample size by using the whole age span, the quality of bootstrapping would most likely be lower for the bins belonging to the tails of the age distribution (e.g., < 75 or > 175) compared to the center bins (e.g., age within $[75, 175]$). In other words, if a bin has too few subjects, an extensive resampling is unlikely to approximate the true distribution of image data for that age bin. Based on the age distribution outlined in Figure R2, the interval $[75 \text{ months}, 175 \text{ months}]$ defined according to FWHM is a compromise between those extremes. The interval encompassed 80% of the subjects and each age bin contained a sufficient number of samples. In other words, this interval resulted in a large \mathbf{y} -conditioned cohort for which accurate bootstrapping in each bin was possible. We now add this point to the Discussion section:

“In the bone age experiment, the interval was selected as the Full Width at Half Maximum (FWHM) [60] of the overall age distribution, which approximately encompassed 80% of the training subjects and focused only on the age range with sufficient samples (Supplement Fig S10). This well-represented age interval facilitated the decorrelation with respect to gender and resulted in a large \mathbf{y} -conditioned cohort for training CP.”

Moreover, we also added several alternative strategies for defining the interval to the Discussion section:

“Another strategy for defining the interval (not explored in this article) is to model the interval as a hyperparameter, whose optimal setting is determined via parameter exploration during nested cross-validation. Alternatively, one can bypass the need of selecting interval by using data-driven matching procedures (e.g., a bipartite graph matching [51] or greedy algorithm [1], which in our experiments produced similar accuracy scores as the one based on the FWHM criteria and bootstrapping.”

(2) “our adversarial loss theoretically achieves statistical mean independence between confounder and the learned features [3]” This claim is very weird for me, because [3] does not give any theoretical guarantee when \mathbf{y} -conditioning techniques are incorporated. Please clarify the sentence.

Response: We rephrase this sentence in the Discussion section.

“This improvement is achieved by our novel loss function based on squared correlation (see Methods section), which encourages statistical mean independence between the derived high-dimensional features and a scalar extraneous variable (in our case, a confounder). When applying this adversarial loss to subjects from the \mathbf{y} -conditioned cohort, CF-Net outperformed other state-of-the-art deep models in learning impartial features and unbiased model interpretation.”

(3) “we used data augmentation to generate two cohorts of equal size and conned the predictions of age by CP to controls (i.e., they-conditioned cohort was defined by $\mathbf{y} = 0$).” (3-1) Why do we need to generate two cohorts of equal size?

Response: We favor balanced training data sets to avoid the risk of the learning model biasing its decision towards the cohort with the larger number of samples. When the sizes of cohorts are unbalanced, one way for the algorithm to lower the training loss is to assign subjects to the larger cohort. To avoid this from happening, a variety approaches have been suggested such as creating cohorts of equal sizes via bootstrapping (our method), weighing the two cohorts so that they are of equal importance within the learning objective function, and sampling equal number of subject from each cohort within each mini-batch. We now briefly mention the rationale of balancing cohorts in Section 2.1:

“On the four folds used for training, two cohorts of equal size were generated by data augmentation (see Methods section) to ensure the model would not bias predictions towards the larger cohort, i.e., the control cohort.”

Note, the augmentation was only applied to the training set. During testing, the prediction accuracy was measured on raw testing subjects, and we used Balanced Accuracy (BAcc) to account for different numbers of testing subjects in each cohort.

“The prediction accuracy on the testing folds was measured by balanced accuracy (BAcc) [40] (to account for different numbers of testing subjects in each cohort...”

(3-2) What data augmentation did you use (for reproducibility) ?

Response: To answer this question, we added a pointer “(see Methods section)” in the Results section:

“On the four folds used for training, two cohorts of equal size were generated by data augmentation (see Methods section) to ensure the model would not bias predictions towards the larger cohort, i.e., the control cohort.”

The Method section provides a detailed description of the data augmentation:

“As in [4], data augmentation produced new synthetic 3D images by randomly shifting each MRI within one voxel and rotating within 1° along the three axes. The augmented dataset included a balanced set of 1,024 MRIs for each group (control and HIV). Assuming that HIV affects the brain bilaterally [43, 1], the left hemisphere was flipped to create a 2nd “right” hemisphere. During testing, the right and “flipped” left hemispheres of the raw test images were given to the trained model, and the prediction score averaged across both hemispheres was used to predict the individual’s diagnosis group.”

Recommendations Below are several recommendations for authors to make a paper strength or clear, while it it not requirements for acceptance.

(3) Methodological perspectives.

(3-1) Baseline methods

The main manuscripts compare the ConvNet and CF-Net, however, the outperforming the convnets is somewhat trivial as suggested by many other invariant feature learning papers. In other words, from the current main manuscripts, the contribution of y-conditioning itself, which is the main methodological contributions, is not clear. On the other hands, the supplemental materials contain several comparisons with more reasonable baselines, which might be more interesting for potential readers. I recommend authors to move some comparison between more reasonable methods into main manuscripts.

Response: While we agree with the reviewer that the comparison is of great interest to readers with a technical background, we left it in the supplement as we do not want to disengage less technical inclined readers of the journal. Fully appreciating the comparison requires in-depth familiarity with the topic of invariant feature learning. To attract both technical and non-technical readers, the main text focuses on the concept and proper use of invariant-feature-learning schemes in medical imaging applications, while the supplement highlights the technical advantage of CF-Net (the squared-correlation-based adversarial loss) over existing methods.

(3-2) Regarding [54]. As, the authors clarified, [54] did not apply the method the case where the confounders are continuous. However, the formulation of [54] itself is general enough for continuous case. This is also clearly mentioned in [54] as below (and as the authors may acknowledge). “Note that under our framework, in theory, s can be any type of data as long as it represents an attribute of x . For example, s can be a real value scalar/vector, which may take many possible values, or a complex sub-structure such as the parse tree of a natural language sentence. But in this paper, we focus mainly on instances where s is a discrete label with multiple choices.”. As the authors mentioned in the response, the log-likelihood of continuous variables is defined by the mean squared error, and the [54] can be easily applied for continuous case. Thereby, the extensive comparison with [54] is possible, and the results might strengthen the paper.

Response: Please note that the baseline “Sadeghi et al. 2019 [7]” in the HIV and NCANDA experiments was an extension of [54] based on the mean squared error (MSE). Therefore, we note in the Supplement Section B:

“First, Sadeghi et al. [7] proposed to use the MSE loss between the predicted and ground-truth confounder value as the adversarial loss for invariant feature learning. Their implementation was confined to the scenario where the prediction network was a logistic regression (linear classifier). To translate that method to our application, we simply replaced the correlation loss of $\mathbb{C}\mathbb{P}$ with the MSE loss. Note, in the binary case, MSE could be replaced with the binary cross-entropy resulting in an implementation that is very similar to the one proposed in [54].”

(4) The impact for medical practitioners.

In my opinion, the potential impact for medical practitioners are not clear from the current manuscripts. For example, when the practitioner should use the proposed method? How can we incorporate the domain knowledge (if possible)? To make the broader impact, such a discussion is preferable.

Response: We now add the following paragraph to the discussion:

“Based on these different \mathbf{y} -conditioning strategies, medical researchers can use CF-Net to train deep models on cohorts not strictly matched with respect to confounders without discarding unmatched samples. However, this does not mean that there is no need to keep the confounders in mind when recruiting participants for medical imaging studies. For all learning models, performing analysis on confounder-matched cohorts with sufficient samples remains a fundamental strategy to disentangle biomarkers of interest from effects of confounders. For example, in the bone age experiment, recruiting enough age-gender-matched samples resulted in a large \mathbf{y} -conditioned cohort that reduce the risk of overfitting during the training of CP. Conversely, if two cohorts have completely different distributions with respect to a confounder (e.g., one has participants with strictly larger age than the other), there is no guarantee that any method, including ours, can remove the bias in a purely data-driven fashion. Therefore, in the study design stage, defining potential confounders for a specific medical application may require domain-specific knowledge to maximize the power of CF-Net in practice.”

Response to Comments from Reviewer #2:

The authors have done a good job responding to the comments from myself and the other reviewers and the manuscript is markedly improved as a result. I have no further comments or suggestions.

Response: We thank the reviewer for the positive feedback.

Response to Comments from Reviewer #3:

Thanks for the detailed response to my previous comments. I have one remaining comment. For the added DeLong tests in Tables 1 and 2, please provide the p -values in the Supplementary materials. The authors used 0.05 as the significance threshold for p -values. However, given the number of tests the authors performed, the authors should correct for multiple hypothesis testing. In Tables 1 and 2, 3 out of 8 tests with p -values slightly less than 0.05 are precisely the scenario where a correction is needed.

Response: As requested, we provided uncorrected p -values of the DeLong’s test in Table S1 and S2 in the supplement. Furthermore, we discuss the outcome of this analysis in the Supplement Section B:

“None of the p -values met the significance threshold after Bonferroni correction ($p < 0.05/3 = 0.017$). However, based on the uncorrected threshold (two-tailed $p < 0.05$), CF-Net resulted in trend-level improvement ($p = 0.069$) in prediction accuracy over ConvNet on the whole cohort. Moreover, only CF-Net resulted in significantly higher accuracy on the \mathbf{c} -independent subset ($p = 0.035$) and on the younger participants from the \mathbf{c} -independent subset ($p = 0.045$).”

Reviewers' Comments:

Reviewer #1:

Remarks to the Author:

Thank you for the detailed comments. All my concerns have been addressed, and I have no further comments and questions.

Reviewer #3:

Remarks to the Author:

Thanks for addressing my previous comments and incorporating the corrected p-values in the Supplement. Preferably, the authors can point out or discuss the results after correction for multiple hypothesis testing in the discussion section of the main text. Besides this, I have no future comments/questions.

Submission ID: NCOMMS-20-23428B
**Training Confounder-Free Deep Learning Models
for Medical Applications**

We address the last comment from the reviewer “Thanks for addressing my previous comments and incorporating the corrected p-values in the Supplement. Preferably, the authors can point out or discuss the results after correction for multiple hypothesis testing in the discussion section of the main text.” by adding the following text in the Discussion section:

“When applying this adversarial loss to subjects from the y -conditioned cohort, CF-Net outperformed other state-of-the-art deep models in classification accuracy. Although this improvement did not meet the significance level after multiple comparison correction, CF-Net resulted in impartial features and unbiased model interpretation according to the experiments in Supplement Sections A through D.”